# Comprehensive genomic and epigenomic analysis in cancer of unknown primary guides molecularly-informed therapies despite heterogeneity

Lino Möhrmann [1,2,3,29], Maximilian Werner [1,29], Małgorzata Oleś [4,5,29], Andreas Mock [5,6,29], Sebastian Uhrig[4,7,29], Arne Jahn [2,3,8], Simon Kreutzfeldt [5], Martina Fröhlich[4,7], Barbara Hutter [4,7], Nagarajan Paramasivam[4,7], Daniela Richter[1,3], Katja Beck[7], Ulrike Winter[7], Katrin Pfütze[7], Christoph E. Heilig [5], Veronica Teleanu[5], Daniel B. Lipka [5,9], Marc Zapatka[10], Dorothea Hanf[1,2], Catrin List[1,2], Michael Allgäuer [11], Roland Penzel[11], Gina Rüter[12], Ivan Jelas[12], Rainer Hamacher [13,14], Johanna Falkenhorst[13,14], Sebastian Wagner[15,16], Christian H. Brandts[15,16], Melanie Boerries [17,18], Anna L. Illert[18,19], Klaus H. Metzeler [20,21], C. Benedikt Westphalen[20], Alexander Desuki[22,23], Thomas Kindler[22,23], Gunnar Folprecht[2], Wilko Weichert[24], Benedikt Brors [25], Albrecht Stenzinger[11], Evelin Schröck[2,3,8], Daniel Hübschmann [4,7,26,27], Peter Horak [5,27], Christoph Heining[1,2,3,30], Stefan Fröhling [5,27,30] & Hanno Glimm [1,2,3,28,30✉]

The benefit of molecularly-informed therapies in cancer of unknown primary (CUP) is unclear. Here, we use comprehensive molecular characterization by whole genome/exome, transcriptome and methylome analysis in 70 CUP patients to reveal substantial mutational heterogeneity with TP53, MUC16, KRAS, LRP1B and CSMD3 being the most frequently mutated known cancer-related genes. The most common fusion partner is FGFR2, the most common focal homozygous deletion affects CDKN2A. 56/70 (80%) patients receive genomics-based treatment recommendations which are applied in 20/56 (36%) cases. Transcriptome and methylome data provide evidence for the underlying entity in 62/70 (89%) cases. Germline analysis reveals five (likely) pathogenic mutations in five patients. Recommended off-label therapies translate into a mean PFS ratio of 3.6 with a median PFS1 of 2.9 months (17 patients) and a median PFS2 of 7.8 months (20 patients). Our data emphasize the clinical value of molecular analysis and underline the need for innovative, mechanism-based clinical trials.

A full list of author affiliations appears at the end of the paper.

Cancer of unknown primary site (CUP) is defined as metastatic cancer without detection of a tumor of origin and accounts for 2–4% of all malignancies. Treatment options are limited and in the majority of cases insufficient[1,2]. Routine diagnostic measures contain blood analyses, histopathology including immunohistochemistry as well as imaging of thorax, abdomen, and pelvis. Even though the prognosis is unfavorable in the majority of cases, ~15% of patients can be categorized into more favorable subsets that benefit from treatment similar to therapies administered to patients with corresponding primary tumors and metastatic spread. Therefore, identification of those favorable CUP subtypes is important for efficient treatment[3]. Currently, CUP subtype identification is mainly based on disease localization and a systematic, in-depth immunohistochemical assessment, but considerable research efforts aim at improving its accuracy or applicability to more patients by new technologies. RNA expression profiling has been reported to identify the tissue of origin in 80–90% of cases[4–7]. Epigenetic profiling using DNA methylation signatures has been reported to predict the tissue of origin in almost 90% of cases[8]. Nevertheless, to date there is no clear evidence from prospective randomized trials that site-specific therapy based on these new approaches leads to improved patient outcome[9,10].

Broad, mainly panel-based, molecular analyses of patients with CUP revealed profound genetic heterogeneity with most frequent alterations in common cancer-related genes like *TP53*, *KRAS*, *CDKN2A*, and *SMAD4*[11–13]. In many cases genetic alterations were identified[14] that potentially can be addressed therapeutically[15]. Still, the clinical benefit of molecularly guided treatment in CUP remains unclear.

Here we describe a cohort of 70 CUP patients characterized by comprehensive molecular profiling within the MASTER program of the National Center for Tumor Diseases and the German Cancer Consortium (NCT/DKTK) combining whole-exome/genome sequencing, transcriptome and methylome analysis in a clinical workflow to identify therapeutic targets. Based on deep insights into the highly individual molecular landscape of the disease, molecular tumor board recommendations led to genomics-guided treatment in 20 patients.

## Results

**Patients.** Seventy CUP patients were included of whom 61 met the criteria defined by the ESMO clinical practice guideline[3]. In the remaining nine cases documentation of necessary initial imaging procedures was lacking (such as CT scans of thorax, abdomen and pelvis). Median age was 46 years (range 18 to 73). 27/70 (39%) patients were male, 43/70 (61%) were female. Median follow-up time was 25.9 months. Median overall survival (OS) was 22.1 months. 38 patients died during the observation period. Documentation of previous therapies and tumor burden was available for 69 patients. The median number of systemic therapies prior to sample submission for these 69 patients was 1 (range 1–7). Detailed patient characteristics are depicted in Table 1 and Supplementary Data 1 and 2.

**Somatic molecular characteristics.** We performed whole-exome sequencing (WES; 41/70, 59%) or whole-genome sequencing (WGS; 29/70, 41%) of tumor DNA and control (germline) DNA derived from peripheral blood. RNA sequencing was performed in 55/70 (79%) cases. Within the coding sequence, 0 to 1418 non-silent point mutations (SNVs, median = 42.5) and 0 to 39 small insertions/deletions (indels, median = 3) were identified per sample (Fig. 1). Three samples had a significantly higher tumor mutational burden (TMB) than all others (≥10 mutations per megabase, Fig. 1). Two-thirds (*n* = 46) of all samples had

### Table 1 Patient characteristics.

| Characteristics | No. of patients |
|---|---|
| All | 70 (100%) |
| **Sex** | |
| Male | 27 (39%) |
| Female | 43 (61%) |
| **Age** | |
| ≤29 | 8 (11%) |
| 30–39 | 10 (14%) |
| 40–49 | 31 (44%) |
| 50–59 | 15 (21%) |
| 60+ | 6 (9%) |
| **Tissue molecular testing method** | |
| WGS | 29 (41%) |
| WES | 41 (59%) |
| RNAseq | 55 (79%) |
| Methylome | 55 (79%) |
| **Histologic diagnosis** | |
| Adenocarcinoma | 43 (61%) |
| Squamous cell carcinoma | 9 (13%) |
| Neuroendocrine tumor | 5 (7%) |
| Melanoma | 3 (4%) |
| Poorly differentiated carcinoma | 3 (4%) |
| Undifferentiated carcinoma | 3 (4%) |
| Carcinoma, not otherwise specified | 2 (3%) |
| Moderately differentiated carcinoma | 1 (1%) |
| Sarcoma | 1 (1%) |
| **Number of metastatic sites** | |
| 1 | 24 (34%) |
| 2 | 20 (29%) |
| 3 | 12 (17%) |
| 4 | 11 (16%) |
| 5 or more | 2 (3%) |
| **Location of disease** | |
| Liver | 39 (56%) |
| Lymph nodes | 36 (51%) |
| Bones | 21 (30%) |
| Lung | 15 (21%) |
| Peritoneum | 14 (20%) |
| Other | 25 (36%) |
| **MTB recommendations** | |
| Therapies recommended | 56 (80%) |
| Therapies applied | 20 (29%) |
| **Number of previous systemic therapies** | |
| 0 | 10 (14%) |
| 1 | 26 (37%) |
| 2 | 17 (24%) |
| 3 | 5 (7%) |
| 4 or more | 11 (16%) |

Cohort description including sex and age distribution, method of molecular analysis and summation of (applied) tumor board recommendations. Number of previous therapies counted until sample submission. Percentages may not total 100 due to rounding.
*WGS* whole-genome sequencing, *WES* whole-exome sequencing, *MTB* molecular tumor board.

alterations (SNVs, indels and fusions) occurring in genes, which were mutated in more than 10% of all samples. Those included nine genes: *TP53*, *TTN*, *MUC16*, *ABCA13*, *COL6A3*, *KRAS*, *LRP1B*, *XIRP2* and *CSMD3*, of which five (*TP53*, *MUC16*, *KRAS*, *LRP1B* and *CSMD3*) are known to be highly relevant in cancer[16]. Three samples (4%) in our cohort harbored a mutation in the *CDKN2A* gene, comprising one stop-gain SNV, one nonsynonymous SNV and one frameshift deletion. Genes most commonly affected by SNVs were *TP53* (*n* = 24), *TTN* (*n* = 19), *MUC16* (*n* = 10), *ABCA13* (*n* = 9), *COL6A3*, *CSMD3*, *LRP1B*, *XIRP2* (*n* = 8 each), *KIAA1109*, *KRAS*, *OBSCN* (*n* = 7 each), *CSMD1*, *DNAH12*, *NEB*, *PCLO* and *PTPRF* (*n* = 6 each). Genes most commonly affected by indels were *ARID1A* (*n* = 3), *APC*,

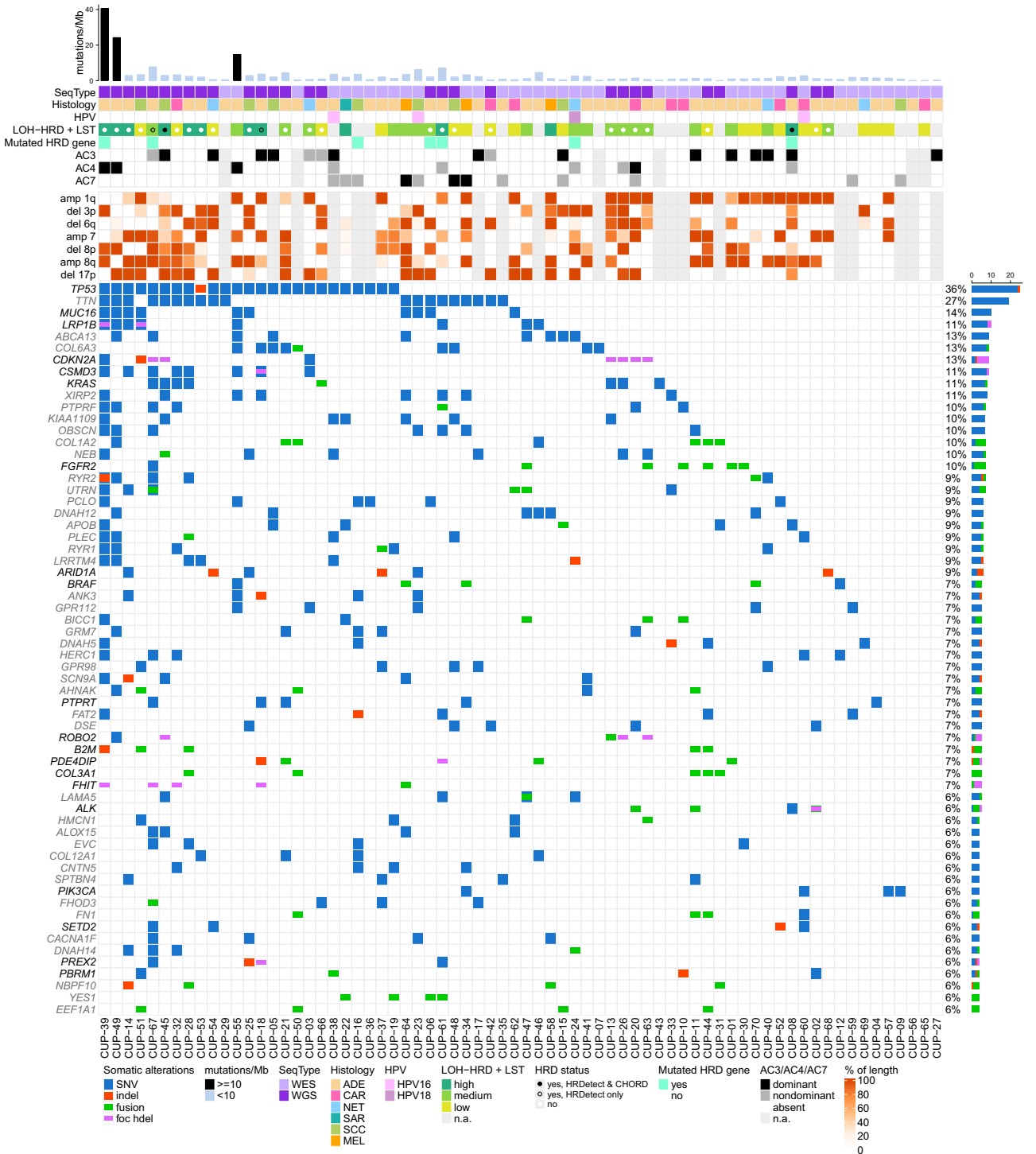

DNAH7, ESRRA, LMTK2 and ZNF107 (n = 2 each). Based on RNA sequencing, we detected 0 to 61 (median = 10, mean = 15) gene fusions of high confidence per patient, many of which were of unclear relevance. In 17 patients we detected fusions with predictive or diagnostic relevance (Supplementary Data 3). The genes that were most commonly involved in these fusions were *FGFR2* (6 patients), *EML4-ALK* (3 patients), *BRAF* (3 patients) and *EWSR1-WT1* (2 patients). In four cases the *FGFR2* fusion was intrachromosomal and included inversions (n = 3, twice with *BICC1*), translocations (n = 2) and a deletion (n = 1). All samples tested for microsatellite instability (MSI) were microsatellite stable (MSS, n = 69). Analysis of somatic copy number aberrations

(sCNA) could be reliably performed in 51 samples (27 WGS and 24 WES). We identified diverse tumor ploidies ranging from two to six (2: n = 31; 3: n = 6; 4: n = 11; 6: n = 3), as well as complex copy number profiles. Furthermore, we found gain and loss events involving single arms or entire chromosomes occurring in multiple samples (Supplementary Fig. 1). In at least 40% of the samples, we detected (i) gains in chromosomes 8q, 1q and 7 (with peaks in loci 8q24.21, 1q42.2, 7q11.21 and presence in 65%, 63% and 55% of samples, respectively) and (ii) losses in chromosomes 6q, 17p, 3p and 8p (with peaks in loci 6q26, 17p13.1, 3p14.2, 8p12 and 8p23.2 and presence in 46%, 46%, 44%, 41% and 41% of samples, respectively). Notably, locus 17p13.1 includes *TP53*. The

**Fig. 1 Molecular alteration landscape and TMB of CUP patient cohort.** Complex characteristics are presented for each patient sample (*x*-axis). The bar plot on top shows the sum of non-silent somatic single-nucleotide variants (SNVs) and coding small insertions/deletions (indels) in exonic sequences per 1 Mb of the coding sequence of the genome. Three samples above 10 mut/Mb threshold (black bars) had a very high mutational burden and were excluded from the pool of mutations considered for the thresholding of genes in the bottom part of the figure. Directly below, the annotation shows: (i) the sequencing type performed, (ii) histology, (iii) HPV infection status, (iv) homologous recombination deficiency (HRD) scores (LOH-HRD + LST), which were defined as high (>20), intermediate (in range 11–20) or low (≤10) together with HRD annotation from HRDetect (*p* > 0.7) and CHORD (p ≥ 0.5), (v) presence of mutation (SNV, indel and fusion of high confidence) in genes related to HRD[75] (Supplementary Data 18), (vi) the presence and dominance status of mutational signatures AC3, AC4 and AC7. The panel underneath shows the percentage (on a continuous color scale from white to orange) of affected genomic fragments by the most frequent copy number events, including amplifications of chromosomes 1q, 7, 8q and deletions of chromosomes 3p, 6q, 8p, and 17p. The genomic coordinates of minimum overlapping fragments with sCNAs occurring at highest frequencies are: 1: 231909967-231965044, 7: 61969019-62050023, 8:128229683-128247675, 8:128340019-128351894, 8:128351920-129149936 for amplifications, and 3:60450070-60453492, 6:162542556-162630329, 8:3094996-3159994, 8:3570000-3590017, 8:3630011-3660004, 8:32400042-32409988, 8:33915011-33989984, 17:9010240-9010378 for deletions. The bottom panel presents the most frequently observed non-silent SNVs (blue), indels (red), fusions of high confidence (green) and focal homozygous deletions (pink; only in genes related to cancer[16]). Only genes mutated in 4 or more patients (while excluding samples with more than 10 mutations/megabase from counting) appear in the plot. Genes in black font color are listed in the Cosmic Cancer Gene Consensus[16], genes in gray font color are not. Source data are provided as a Source Data file.

most common focal copy number aberrations identified in WGS samples included: (i) amplification involving a fragment of 8q24.21 and affecting *MYC* (*n* = 4), (ii) homozygous deletion involving a fragment of 22q12.2 affecting *TTC28* (*n* = 7)—which inhibits tumor cell growth by interacting with *TP53*[17], and (iii) deletion involving a fragment of 9p21.3 directly affecting tumor suppressor *CDKN2A* (*n* = 7; homozygous deletion in six samples, and deletion of one copy in one sample). Homologous recombination deficiency (HRD) could be assessed in 51/70 patients by calculating a score (sum of LOH-HRD and LST). The score was high (>20) in twelve, intermediate (in range 11–20) in 19 and low (≤10) in 20 patients. Additionally, we used two WGS-based methods, HRDetect and CHORD, which produced a software-specific HRD probability score. The probability was >0.99 and >0.93, respectively, in two patients (CUP-45 and CUP-08). Both of them were the highest-scoring WGS samples of the first method (sum of LOH-HRD and LST), which was independent from sequencing type (Supplementary Fig. 2). Analysis of the sequencing data for reads of viral origin revealed three tumors, which were positive for human papillomavirus type 16 and another one, which was positive for human papillomavirus type 18 (Supplementary Table 1). Integration of viral DNA into the host genome was detected in three cases inside/near the genes *HOXA2*, *MIER1*, *SLC35D1*, and *TENM4*. Mutational signature analysis revealed signatures associated with impaired homologous recombination (AC3) in seventeen, with UV exposure (AC7) in twelve, with tobacco smoking (AC4) in nine, and with alkylating agent exposure (AC11) in five samples. These signatures were dominant in thirteen, three, four, and zero samples, respectively (Supplementary Fig. 3). Mutations of *TP53* (27 SNVs, one frameshift insertion) and *KRAS* (seven SNVs) were significantly enriched within the cohort (MutSigCV v.1.4, q < 0.15) with a median allele frequency of 0.42 for *TP53* and 0.48 for *KRAS*[18]. In all but one TP53-mutated samples, sCNA were identified comprising loss of heterozygosity (LOH) and/or deletion (for more details see Supplementary Results).

**Germline analysis**. Assessment of germline variants in all 70 patients revealed five pathogenic or likely pathogenic variants (Supplementary Data 4) in five patients (7%) in the genes *CHEK2*, *BRCA1*, *CDKN2A*, *NBN* and *ERCC3*. Genetic counseling was recommended for the index patients and for their relatives. The mean age at onset of the five patients with pathogenic or likely pathogenic variants was significantly lower than the age at onset of the other 65 patients (29.7 vs. 45.5 years, *p* = 0.001). In addition to the recommendation for genetic counseling, three variants likely associated with an autosomal dominant cancer disposition

supported a treatment recommendation, twice for PARP inhibition (*BRCA1*, *CHEK2*) and once for CDK4/6 inhibition (*CDKN2A*). None of the patients with documented previous tumors (five patients) had a (likely) pathogenic germline mutation. Moreover, loss-of-heterozygosity (LOH) occurred only in one of the patients with a pathogenic variant (*CDKN2A*, CUP-64).

**Entity prediction using methylome and transcriptome data**. We retrospectively performed entity predictions using methylome- and transcriptome-based similarity analysis. For this purpose, we first used methylome and transcriptome data from 33 different TCGA cohorts. We set up a validation cohort using 100 consecutive MASTER patients enrolled between 12/2020 and 06/2021 (Supplementary Data 5) consisting only of entities that are part of TCGA to ensure comparability with other tissue-of-origin classifiers, which were usually trained on TCGA data and can therefore only reliably predict entities that are part of TCGA. Transcriptome data was available for 72 patients of the validation cohort (Supplementary Data 6), methylome data for 77 (Supplementary Data 7). We compared the accuracy of our results (expression comparison TCGA 41/72, 57%; methylome comparison TCGA 52/77, 68%) with two other published methods, cancerSCOPE[19] and CUP-AI-Dx[20], both of which had similar accuracy as our entity prediction when using TCGA as a reference cohort (cancerSCOPE highest score 40/72, 56%; cancerSCOPE consensus 35/72, 49%; CUP-AI-Dx 39/72, 54%). When comparing TCGA cohorts with our CUP cohort, classification based on methylome comparison was possible in 55/70 cases (79%). The remaining cases had a tumor cell content of 30% or less or did not have enough DNA material available for analysis. Classification based on transcriptome was possible in 55/70 cases (79%) as well, but not in the very same 55 patients classified by methylome analysis. The remaining cases had low RNA quality or did not have RNA material available for analysis. In 62/70 cases (89%) entity prediction was possible by at least one of the methods. In 48/70 cases (69%) entity prediction was possible by both methylome and transcriptome analysis. In only 20/48 (42%) the same entity was predicted by both methods, causing us to consider additional data like indicative molecular alterations and histology (Supplementary Data 8). To address this problem, we performed a similarity analysis by comparing the transcriptome data of the entire MASTER cohort across histologies (reference cohort consisting of 1890 samples from 1814 patients, Supplementary Data 9) with our CUP cohort. This approach showed the best accuracy when tested on our validation cohort (56/72, 78%, Supplementary Data 6) and enabled us to have a comparison to rare tumor entities, which were enrolled in MASTER but are not

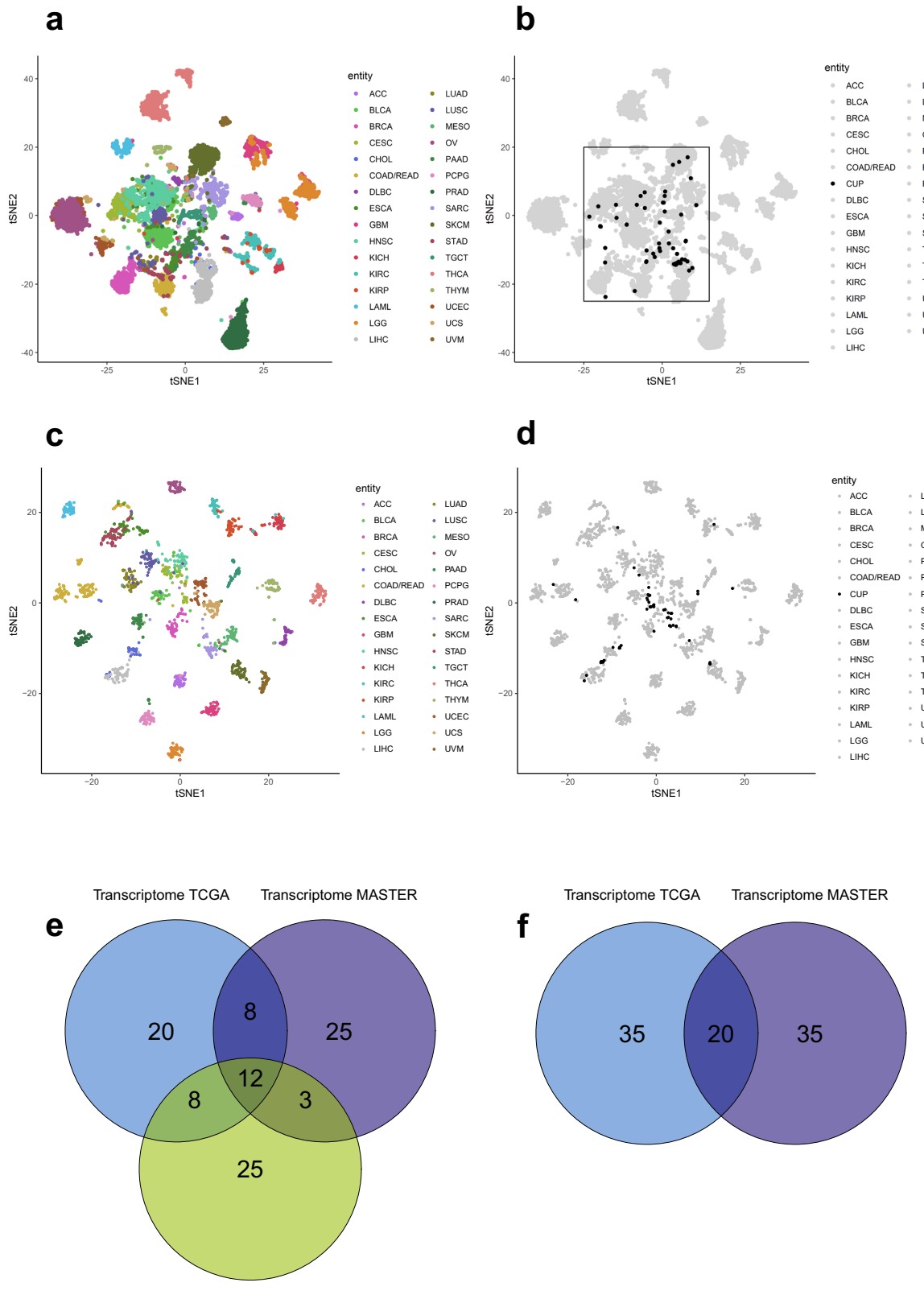

part of TCGA. When plotted graphically (Fig. 2a–d), some TCGA entities showed distinctive clusters, some did overlap with other entities. Cholangiocarcinoma (CHOL) and pancreatic adeno-carcinoma (PAAD) are two examples of entities that were hard to distinguish while at the same time occurring quite often as pre-dictions in our CUP cohort. CUP samples were close to many entities that did not necessarily cluster distinctively. For example, none of the CUPs were close to pheochromocytoma and para-ganglioma (PCPG), which cluster separately from most other entities, but many CUPs were close to gastrointestinal tumors. Taken together, our multi-omics approach led to a higher per-centage of predictions but at the same time it did not clarify the

**Fig. 2 Methylome and transcriptome-based clustering. a** tSNE plot based on the 5000 most variant CpG sites across the TCGA pan-cancer cohort (*n* = 8024, 33 different cancer entities in 32 entity baskets). The tSNE analysis was jointly performed on the complete TCGA and MASTER CUP samples (*n* = 55) to ensure comparability within the landscape. This subplot shows TCGA samples only. While many TCGA entities show distinctive clusters, some do overlap with other entities. **b** This subplot illustrates all MASTER CUP patients (black) on top of the TCGA sample landscape (gray). CUP samples were close to many entities that did not necessarily cluster distinctively (e.g., gastrointestinal tumors). **c** Transcriptome-based tSNE clustering of 33 different cancer entities in 32 baskets using TCGA data without CUP patients (*n* = 1809). **d** Transcriptome-based tSNE clustering of MASTER CUP patients (black, *n* = 55) among the background of TCGA-based clusters (gray). As with the methylome-based clustering, a notable fraction of the samples are found in the diffusely structured center of the tSNE clustering. **e** Venn diagram depicting concurring results between methylome-based CUP entity predictions (comparison to TCGA) and transcriptome-based entity predictions (comparison both to TCGA and MASTER; each depicted in separate groups). 48 patients of the CUP cohort had predictions based on all three methods and were therefore eligible for comparison (Supplementary Data 8). **f** Venn diagram depicting concurring results between both transcriptome-based CUP entity predictions (TCGA and MASTER comparison). 55 patients of the CUP cohort had transcriptome-based predictions (Supplementary Data 8). Source data are provided as a Source Data file.

diagnosis in a subgroup of patients (concurring and discrepant results presented in Venn diagrams, Fig. 2e, f). It highlights the need for an integrated classifier taking into account both methylation and transcriptomic data (detailed information on entity predictions in Supplementary Data 8).

**Diagnostic interpretation of molecular profiling**. We detected several rare genetic alterations of diagnostic interest (Supplementary Data 8). We found an *IDH1* p.R132C mutation (CUP-68) and an *IDH2* p.R172G mutation (CUP-19), which have been reported to occur in a subtype of cholangiocarcinoma[21,22]. Both offer a rationale for treatment with IDH inhibitors. In CUP-03 we detected a *KIT* p.V560D mutation and transcriptomic profiling matched with thymic carcinomas. Although *KIT* mutations occur in different entities, they have been described to occur in thymic carcinomas[23] and can be addressed therapeutically by several generations of tyrosine kinase inhibitors.

Additionally, we were able to identify several oncogenic gene fusions of interest (Supplementary Data 3). In six patients, we detected a gene fusion involving *FGFR2*, which always occurred at the same splice-site of the *FGFR2* gene (Fig. 3a). These fusions are typical for a subgroup of cholangiocarcinoma[24]. Usually, the intact kinase domain of *FGFR2* is fused with a gene, which provides a dimerization/oligomerization domain that facilitates constitutive activation of downstream RAS/MAPK and PI3K/AKT pathways. Moreover, they can be addressed therapeutically using FGFR inhibitors[25]. *EML4-ALK* fusions were detected in the tumors of three patients (3/70; CUP-02, CUP-20, CUP-11) and could be confirmed by immunohistochemistry in all cases. At the same time CUP-33 was reported to have a previously detected *EML4-ALK* fusion that could not be found in our analysis, most likely due to low quality of the biopsy sample. *EML4-ALK* fusions are characteristic for a subtype of non-small-cell lung cancer (NSCLC) and can be addressed therapeutically using ALK inhibitors[26]. We detected two *EWSR1-WT1* fusions (CUP-15, CUP-42), which are typical for desmoplastic small round cell tumors[27]. Two patients diagnosed with melanoma of unknown primary by histologic analysis (CUP-34 and CUP-64) harbored two *BRAF* fusions each, of which CUP-34 also had a *BRAF* V600E mutation.

CUP-14 had an *ARHGAP26-CLDN18* fusion, which has been described in gastric signet-ring cell carcinoma[28]. A previously undescribed *MXI1-NUTM1* fusion was detected and confirmed by routine diagnostics in CUP-04 (Fig. 3b). *NUTM1* fusions define NUT midline carcinomas[29] and usually involve *BRD3* or *BRD4* as fusion partners. In CUP-08 we detected an *EZR-ERBB4* fusion, which has been described only once in *KRAS* wild type invasive mucinous lung adenocarcinoma[30]. Furthermore, we identified an *ITSN1-BRAF* fusion in CUP-70 that has not been described yet. However, without additional information the

fusion event can't be linked to a certain entity since *BRAF* fusions play a role in various tumor entities[31].

In total, 20/70 (29%) patients harbored rare genetic alterations that could be linked to specific entities. Not all of them correlated with entity prediction results based on transcriptome profiling and these alterations alone did not necessarily justify diagnostic reclassification. Still, together with omics-based entity predictions they offer meaningful information of diagnostic value and can be useful for further treatment decisions.

**Genomics-based treatment recommendations**. A dedicated molecular tumor board (MTB) recommended personalized therapy options based on information from DNA and RNA sequencing for 56/70 (80%) patients in our cohort. Median turnaround time from sample submission to MTB was 2.3 months (range 0.7–7.0 months). In four cases without recommendation (4/70) the tumor cell content of the respective biopsy was not sufficient for analysis, in six cases (6/70) the respective patients died before the tumor board could convene, in two cases (2/70) the sample quality was not sufficient for analysis and in the remaining two cases (2/70) no targetable mutations were found. Two patients received a second MTB recommendation (MTBR) based on a follow-up biopsy after progression of disease occurred while being treated with molecularly guided therapy (pazopanib, CUP-42; cetuximab + carboplatin + paclitaxel, CUP-70). In total, 58 MTBRs were issued. The first 56 MTBRs contained 142 drug recommendations, which were grouped into eight different baskets (tyrosine kinases, PI3K-AKT-mTOR, RAF-MEK-ERK, developmental pathways, DNA damage response, cell cycle regulation, immune evasion and others) by the type of pathway a recommended drug interacts with[32]. Tyrosine kinases were the most common basket used for recommendations (47/164, Fig. 4a and Supplementary Data 10). All drug recommendations were sorted into groups by the NCT/DKTK evidence level they were based on as described by Leichsenring and colleagues[33] (Level 1A/B/C, 11/142, 8%; Level 2A/B/C, 89/142, 63%; Level 3, 31/142, 22%; Level 4: 11/142, 8%). 18 samples had a mutational burden of at least 100 non-silent SNVs and coding indels, which was defined as hypermutation by the MTB and used as a potential rationale for immune checkpoint inhibitor recommendations (immune evasion basket).

**Genomics-based systemic treatment**. Twenty (20/56, 36%) patients were treated in accordance with MTBRs using 30 applied drugs or drug combinations (Fig. 5 and Supplementary Data 11). Treatments were given off-label at the discretion of the treating oncologist. Recommendations based on the immune evasion basket were most likely to be applied (Fig. 4b). The distribution of NCT/DKTK evidence levels of the first clinically applied recommendations showed a similar distribution as the

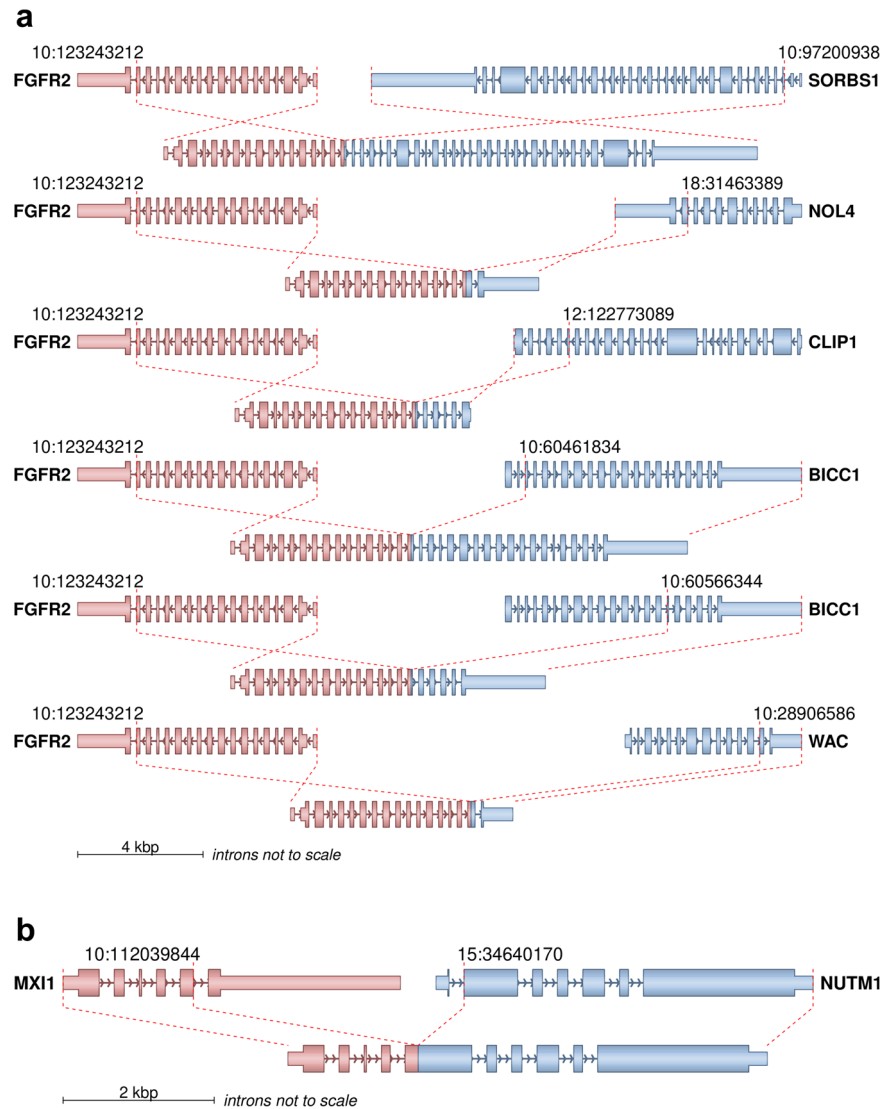

**Fig. 3 Fusions of high confidence.** Exon structures in transcriptome sequencing data are shown. **a** Fusions involving *FGFR2* found in six patients. Fusion partners were *CLIP1, NOL4, WAC, SORBS1* and twice *BICC1*. **b** Fusion involving *NUTM1* found in one patient. *MXI1* has not been described previously as a fusion partner for *NUTM1*, but *NUTM1* fusions do define NUT midline carcinomas. Therefore, we identify *MXI1* as a new possible fusion partner in NUT midline carcinoma. Source data are provided as a Source Data file.

one of all drug recommendations (Level 1, 2/20, 10%; Level 2, 15/20, 75%; Level 3, 1/20, 5%; Level 4, 2/20, 10%). To evaluate clinical benefit, we calculated the ratio (PFSr) of the progression-free survival time associated with the first applied therapy recommended by MASTER (PFS2) and the progression-free survival associated with the last prior systemic therapy (PFS1). Three (3/20, 15%) patients did not progress during their last systemic therapy before they received a recommended therapy. Therefore, neither PFS1 nor PFSr could be determined for them, but recommended therapies translated into a PFS2 of 6.0, 10.0 and 11.1 months, respectively. Mean PFSr for the other 17 patients was 3.6. Median PFSr was 2.3 with a range from 0.2 to 16.4. Median PFS1 was 2.9 months ($n = 17$) and median PFS2 was 7.8 months ($n = 20$, Supplementary Fig. 4). In one patient the PFS defining event was death, in the others it was progressive disease. To improve concordance with physician-perceived clinical benefit, we calculated the modified PFS2/PFS1 ratio (mPFSr) as described by Mock, Heilig and colleagues[34], which resulted in a mean mPFSr of 5.0 (median mPFSr = 2.7; range 0.2 to 12.0; Table 2).

13/17 treated patients had a mean PFSr > 1.3, which was considered as clinical benefit since this value has been frequently used as a measure for positive clinical outcome in precision oncology trials[35–37]. Four of them received treatment with immune checkpoint inhibitors, three with ALK inhibitors, three with multikinase inhibitors, one with trastuzumab, one with vismodegib and one with olaparib plus gemcitabine (Supplementary Data 12). Three patients had a PFS2 > 1 year, namely CUP-57 (26.5 months, trastuzumab), CUP-18 (23.3 months, nivolumab) and CUP-08 (12.2 months, olaparib plus gemcitabine).

Observed responses were complete response (CR, 1/30, 3%), partial response (PR, 9/30, 30%), stable disease (SD, 8/30, 27%), mixed response (MR, 3/30, 10%) and progressive disease (PD, 9/30, 30%). Of 20 patients who received the recommended targeted therapies, 12 (60%) had stable disease ≥6 months or achieved objective remissions (PR/CR). PR was achieved with the following treatments: crizotinib, CUP-02; olaparib + gemcitabine, CUP-08; pazopanib, CUP-15; nivolumab, CUP-18/CUP-25/CUP-49; bicalutamide + leuprorelin + nab-paclitaxel, CUP-57; trastuzumab, CUP-57; nintedanib + docetaxel, CUP-62.

## Basket distribution

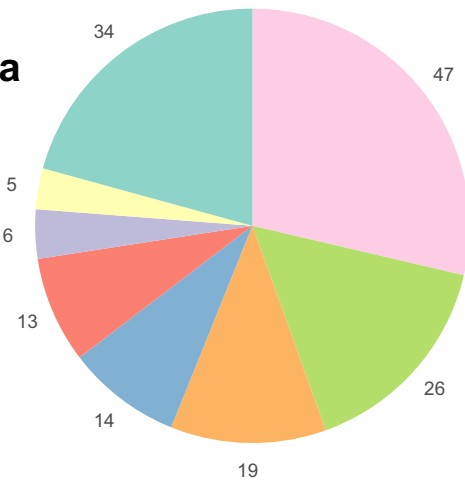

No. of recommended therapies

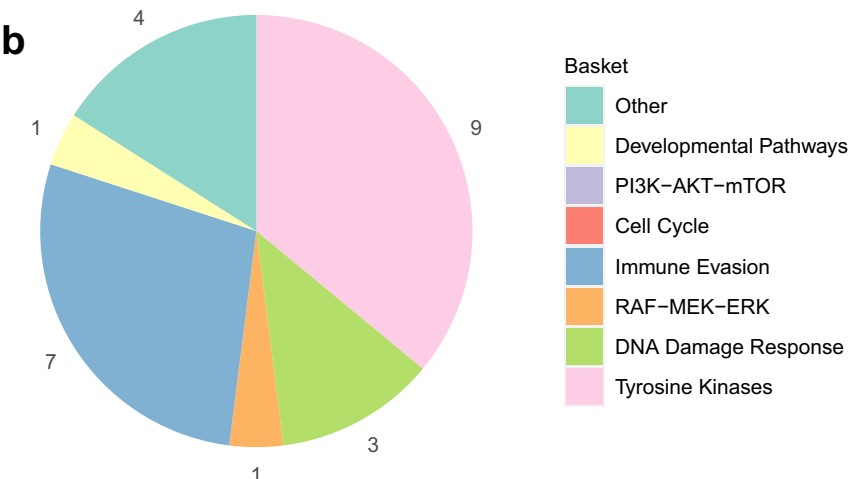

No. of applied therapies

**Fig. 4 Recommended and applied therapies.** Distribution of all therapy recommendations among 8 different baskets. Please note that most patients received several recommendations. Distribution of **a** all therapy recommendations and **b** first 20 clinically applied therapy recommendations among eight baskets: Tyrosine Kinases, DNA Damage Response, Immune Evasion, Cell Cycle, RAF-MEK-ERK, PI3K-AKT-mTOR, Developmental Pathways and Other. Combination therapies were sorted into multiple baskets based on their mechanism of action. Source data are provided as a Source Data file.

Four patients were started on monotherapy or bridging therapies and switched to combination therapy later on (CUP-21, CUP-39, CUP-51, and CUP-57; Supplementary Results). Notably, CUP-51 was started on nivolumab and showed MR so that the subsequent treatment was escalated to a combination of nivolumab and ipilimumab, which led to CR with no reported progression until the end of the observation period (PFS2 9.8 months).

Five patients who received an MTBR were treated with at least one subsequent treatment afterwards, three of them received therapies based on two or more MTBRs, the other two (CUP-38, CUP-70) received chemotherapies. CUP-69 had PD within three months after treatment initiation with the first MTBR (olaparib, PFS2 3.6 months) and was subsequently treated with chemotherapy again (FOLFOX, PFS3 2.9 months). After further progression, a second MTBR was applied (cabozantinib, PFS4

9.4 months). CUP-20 received crizotinib (PFS2 5.9 months) and after progression alectinib (PFS3 6.0 months). CUP-02 received treatment with the ALK inhibitors crizotinib (PFS2 5.6 months), ceritinib (PFS3 10.2 months), alectinib (PFS4: 3.2 months) and brigatinib plus chemotherapy (PFS5 1.7 months) in accordance with the patients' MTBR.

For patients that did not receive a recommended treatment, we calculated the ratio of the first treatment applied after the MTB (PFSb; median = 3.8 months, $n = 12$) and the last prior systemic treatment (PFSa; median = 4.8 months, $n = 11$), which resulted in a mean PFSr of 0.67 (median PFSr = 0.71, range 0.1 to 1.0, $n = 11$; Supplementary Data 13). Median overall survival of the 36 patients without application of recommended treatments was significantly shorter than of the 20 patients that received a recommended therapy (18.3 months vs. 34.8 months, $p = 0.022$). Same was true for median PFS2 and PFSb (7.8 months vs.

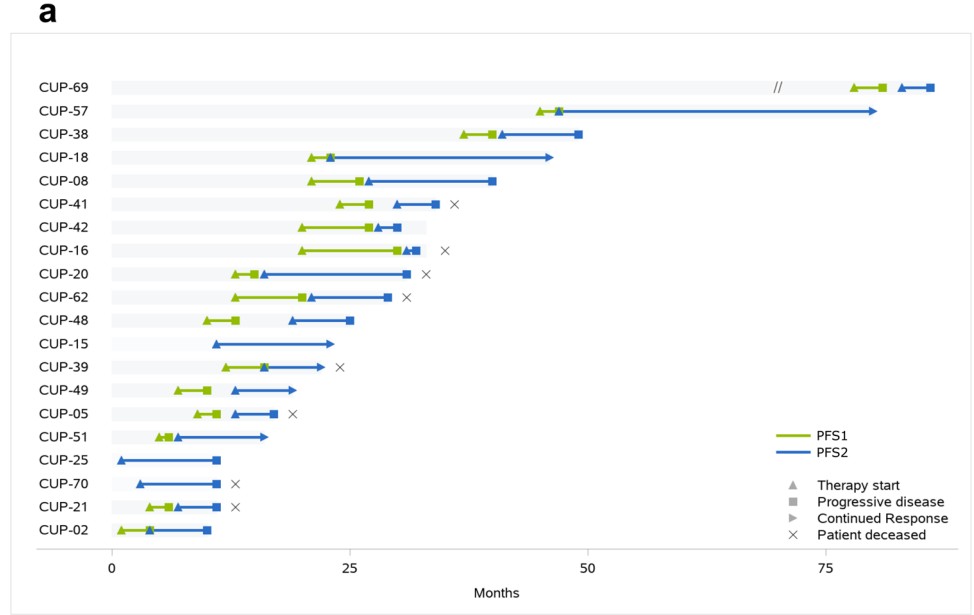

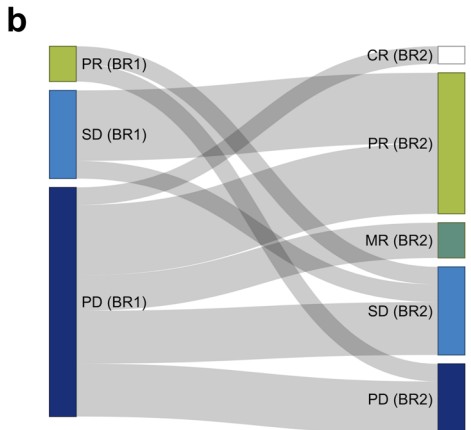

**Fig. 5 Clinical course of 20 patients with molecularly guided therapy. a** Each bar represents one patient in the study starting from date of diagnosis. PFS of the last systemic therapy before molecular analysis (PFS1, green) and the first applied molecularly guided therapy (PFS2, blue) are plotted inside those bars. Continued response at the end of the observation period is marked with an arrow. The CUP-69 bar has been shortened by 60 months for visibility (true length 146 months). For CUP-15, CUP-25 and CUP-70 neither PFSr nor mPFSr could be calculated. **b** Sankey plot depicting best response associated with the last systemic therapy before genomic and transcriptomic analysis (BR1, n = 20) in comparison with best response associated with the first applied molecularly guided therapy (BR2, n = 20). Source data are provided as a Source Data file.

3.8 months, $p < 0.0001$). Two patients with PFSb had stable disease ≥6 months or achieved objective remissions (PR/CR; Supplementary Fig. 4). Since our study was not randomized, these results are not controlled for possible confounding factors. Within MASTER, reasons for non-implementation of MTBRs included lack of availability or reimbursement of recommended treatments, deterioration of a patient's general condition and death before treatment application[38]. Further data are provided in Supplementary Data 13 and 14.

The median number of prior systemic palliative therapies that patients with an applied MTBR had received was three (range 1−7). Eleven had already been treated with targeted therapies indicating that clinical benefit may be achieved even in heavily pretreated patients.

**Discussion**

In addition to known recurrent mutations in CUP, our comprehensive WGS/WES approach detected a variety of rare genetic alterations, which were relevant for molecularly targeted treatment decisions. This suggests that comprehensive molecular analysis is particularly well-suited for this heterogeneous disease.

On the genomic level, we observed frequent mutations in well-known cancer-related genes such as *TP53*, *MUC16* and *KRAS*. The majority of these common alterations has previously been described in studies using gene panel sequencing. When using a 50 gene panel, Löffler and colleagues described *TP53*, *KRAS*, *CDKN2A*, and *SMAD4* as the most frequently mutated genes and *CDKN2A* as the most frequently deleted gene in CUP[11]. Varghese and colleagues reported a variety of targetable alterations in 45/150 CUP patients using MSK-IMPACT, a deep-coverage hybridization capture-based assay encompassing 341 (later expanded to 410) cancer-associated genes accompanied by WES in 13 cases[14]. The most commonly mutated genes were *TP53*, *KRAS*, *CDKN2A*, *KEAP1*, and *SMARCA4* and 15/150 patients received targeted therapies[14]. These common mutations are involved in a variety of cell processes and do not offer a clear rationale for targeted therapies available at the moment.

**Table 2 Clinical outcome.**

|  | Value | n |
|---|---|---|
| PFS1 |  |  |
| Median (range) | 2.9 (1.0–10.0) months | 17 |
| PFS2 |  |  |
| Median (range) | 7.8 (1.6–26.5) months | 20 |
| PFSr |  |  |
| Median (range) | 2.3 (0.2−16.4) | 17 |
| Mean | 3.6 | 17 |
| mPFSr |  |  |
| Median (range) | 2.7 (0.2−12.0) | 17 |
| Mean | 5.0 | 17 |

Overview including PFS1, PFS2, PFSr, mPFSr, depicting respective median and mean values. For three patients, PFS1 could not be determined since there was no progression reported. Modified PFS ratio (mPFSr), prePFS and postPFS were calculated as described by Mock et al.[34] *PFS1* progression-free survival time 1 (last systemic treatment prior to application of a recommended therapy), *PFS2* progression-free survival time 2 (recommended therapy), *PFSr* PFS ratio.

Furthermore, we and others report a substantial amount of pathogenic germline mutations amongst CUP patients[39–42]. These can have a direct impact on the patient and their families. Therefore, germline testing should be considered, especially for young patients with CUP or patients with previous malignancies.

On the transcriptomic level, we detected a variety of therapeutically relevant gene fusions. In addition, we classified tumors based on transcriptomic and epigenetic information, which was complemented by specific disease-defining alterations and by presence of certain dominant mutational signatures. Nevertheless, only a third of the entity predictions based on methylome and transcriptome did match, which may be explained by tumor cell content, RNA quality, differences between TCGA and the MASTER cohort composition, as well as differences in the methods used. The question whether site-specific therapies are beneficial for CUP patients is still a matter of ongoing debate. Using a 92-gene RT-PCR cancer classification assay, Hainsworth and colleagues reported that site-specific therapy leads to significantly improved survival when clinically more responsive tumor types were predicted[9]. In contrast, Hayashi and colleagues reported that site-specific treatment based on microarray profiling did not result in a significant improvement in 1-year survival compared with empirical paclitaxel and carboplatin, although prediction of the original site seemed to be of prognostic value[43]. Similarly, a meta-analysis by Rassy et al. showed no significant survival benefit with site-specific in comparison to empiric chemotherapy. At the same time the heterogeneity across the available data demonstrates that further well-designed trials are needed[10] and that a reliable classification method for the attribution of a CUP case to a specific tumor entity is still to be developed. Moran and colleagues used microarray DNA methylation signatures (EPICUP) to predict a primary cancer of origin in 188 (87%) of 216 CUP patients. In this study, patients with EPICUP diagnoses who received a tumor type-specific therapy showed improved overall survival compared with that in patients who received empiric therapy[8]. Prospective validation of this epigenetic approach is still missing.

In our study, we used epigenetic and transcriptomic analyses together with evaluation of disease-specific mutations to identify the tissue of origin. On the one hand this increased the total number of patients for whom an entity prediction was possible, on the other hand it led to contradictory results in a majority of patients that had several layers of information available. In these cases, it is not clear which data is to be trusted more. Probably, several factors contribute to prediction errors by one or the other method: First, the composition of the MASTER CUP cohort and the TCGA reference cohort differ in several aspects, including metastatic status and represented entity subtypes. Second, differences in the sample preparation protocols between MASTER and TCGA may introduce technical biases, confounding the algorithms of the classifiers. And third, some entities are hard to distinguish using epigenetic/transcriptomic information. For example, the classifiers frequently produced discrepant results concerning hepatic metastases of pancreatic (PAAD) or cholangiocellular (CHOL) carcinoma, which were classified inconsistently as one of PAAD, CHOL or LIHC (liver hepatocellular carcinoma). It is unclear whether the missing success of prospective trials using site-specific treatment is due to limited accuracy in identifying the tissue of origin or due to limited relevance of the site of origin for clinical outcome in the majority of CUP patients. In our cohort, MTB recommendations were not influenced by our methylome- and transcriptome-based entity prediction since it was not available at the time of the MTB. Future trials might benefit from a well-designed integrated classifier taking into account both methylation and transcriptomic data. As previously shown in pancreatic cancer and other hard-to-treat entities, comprehensive molecular profiling also offers the opportunity to detect rare or previously unknown therapeutic targets[44,45]. Therefore, in the light of continuously improving options regarding molecular diagnostics and targeted therapies, genomics-based treatment might be the more promising approach.

Our study had several potential limitations. First, our patient population was younger than one would expect for a representative CUP cohort, which can at least be partially explained by the NCT/DKTK MASTER inclusion criteria. Second, our cohort was treated with a wide range of different therapies prior to molecular analysis. Third, our study was not a randomized clinical trial but a prospective observational study. However, the mean PFS2/PFS1 ratio in our cohort was 3.6. 13/17 treated patients (77%) for which a PFS ratio could be determined achieved a ratio higher than 1.3, the originally proposed threshold for assessment of clinical benefit in previous studies like MOS-CATO 01[35]. The median overall survival in our cohort was significantly longer when compared to published data, which may be partially attributed to the young patient age in our cohort. Nevertheless, our results provide evidence that a considerable part of CUP patients may benefit from comprehensive molecular analysis. Although there are case reports about successful use of checkpoint inhibitors in CUP patients[46], immunotherapy in CUP has not been clinically implemented yet, unless microsatellite instability or DNA mismatch repair (MMR) deficiency have been detected. Our results underline that immunotherapeutic approaches can be efficient in a much larger proportion of CUP patients. Furthermore, we observed a meaningful proportion of CUP patients benefiting from molecularly stratified treatments. Two prospective randomized phase II trials testing novel strategies versus empirical chemotherapy are currently ongoing (CUPISCO, NCT03498521 and CheCUP, NCT04131621).

In conclusion, our findings indicate that comprehensive molecular analysis of CUP patients can be highly beneficial even at late stages or following several rounds of prior treatment.

We provide valuable insight into the heterogenic genomic, transcriptomic and epigenetic landscape of CUP and show potentially actionable alterations in a large proportion of patients. Further prospective clinical studies to assess the impact of genomics-based personalized cancer therapy are warranted.

## Methods

**Clinical and statistical analysis.** The study included 70 patients who were enrolled in the National Center for Tumor Diseases and German Cancer Consortium (NCT/DKTK) Molecularly Aided Stratification for Tumor Eradication

Research (MASTER) precision oncology program between May 2013 and July 2018 with follow-up until June 2019[32,47]. All 70 patients had a CUP diagnosis according to their referring oncologists. Clinical data for cohort selection, description and analysis was obtained from the National Center for Tumor Diseases (NCT) Heidelberg and Dresden, as well as from six other comprehensive cancer centers (CCCs) of the German Cancer Consortium (DKTK). The DKTK network includes ten CCCs at eight sites (Berlin, Dresden, Essen/Düsseldorf, Frankfurt/Mainz, Freiburg, Heidelberg, Munich, Tübingen). Demographic data, histopathological diagnosis, location of metastases at the time of enrollment, fulfillment of the ESMO CUP diagnostic criteria, systemic therapies and staging information, genomic information available at the time of the molecular tumor board (MTB), recommendations of the MTB and application of recommended therapies were assessed and documented in a centrally managed electronic data capture system (ONKOSTAR). MTB recommendations were based on the information obtained from DNA and RNA sequencing. Therapeutic options steadily improved over time. Every tumor board recommendation contains several drugs or drug combinations with different priorities assigned. For our analysis, we included only the first three priorities since there was no drug with a lower priority clinically applied. In some cases, there were fewer than three drugs recommended. All drug recommendations were issued with an NCT/DKTK evidence level reflecting the origin of the information that the respective recommendation was based on[33]. Overall survival (OS) was defined as the time from the date of diagnosis to the date of death or last follow-up. Progression-free survival (PFS) was defined as the time from the date of systemic therapy initiation to the date of death, progressive disease or last follow-up. Median OS, PFS and follow-up time were estimated using the Kaplan–Meier method, and a log-rank test was used to compare OS and PFS among patient subgroups. PFS of the first applied treatment recommended by the MTB (PFS2) was compared to the PFS of the last prior systemic treatment (PFS1) in each individual patient. If more than one recommended therapy was applied, PFS3 and following were calculated. PFS defining events were progressive disease or death, determined by a medical oncologist via review and assessment of the corresponding medical documents. The progression-free survival time ratios (PFSr) between PFS2 and PFS1 were calculated. Modified progression-free survival time ratios (mPFSr) were calculated following the proposal of Mock and colleagues[34]. For patients that did not receive a recommended treatment, we calculated the ratio of the first treatment applied after the MTB (PFSb) and the last prior systemic treatment (PFSa).

**NCT/DKTK MASTER**. NCT/DKTK MASTER is a prospective, continuously recruiting, multicenter observational study that provides a standardized diagnostic workflow, which enables molecularly informed decisions for further therapy. Treatment recommendations are made in cooperation with treating oncologists following interdisciplinary discussion in a molecular tumor board. MASTER includes adults with advanced cancer across all entities who are younger than 51 years and patients with rare tumors, including rare subtypes of more common entities, regardless of age. Patients must have exhausted curative treatment options and be in good general condition (Eastern Cooperative Oncology Group performance status of 0 or 1)[38]. Patients with cancers of unknown primary were included regardless of age due to its rarity.

Patients provided written informed consent for banking of tumor and control tissue, molecular analysis including germline analysis, and the collection of clinical data under a protocol (S-206/2011) approved by the Ethics Committee of the Medical Faculty of Heidelberg University. The study was conducted in accordance with the Declaration of Helsinki. Patients did not receive participant compensation. Molecularly informed therapies were not part of MASTER but given off-label at the discretion of and by the treating oncologist who obtained informed consent for each therapy. German regulations for off-label treatment allow individual treatment decisions after obtaining informed consent and no IRB approval is required. Costs for off-label drugs can be reimbursed by German health insurances if the patient has a severe disease, if there is no other treatment option available and if there is reasonable hope for treatment success based on available scientific or clinical data.

**Entity prediction validation cohort**. We used 100 consecutive MASTER patients enrolled between 12/2020 and 06/2021 (Supplementary Data 5) consisting only of entities that are part of TCGA as a cohort to validate all entity prediction methods and measured their accuracy before using them for CUP entity predictions. Transcriptome data was available for 72 patients of the validation cohort (Supplementary Data 6) methylome data for 77 (Supplementary Data 7).

**Next-generation sequencing and bioinformatic processing**
*Sample preparation and sequencing*. DNA from fresh frozen tumor tissue was isolated using the Allprep DNA/RNA/miRNA Universal Kit (Qiagen) or QIAamp DNA mini (QIAGEN). DNA from formalin fixed paraffin embedded tissue was isolated using the GeneRead DNA FFPE Kit (QIAGEN). DNA from peripheral blood was isolated using QIAamp DNA Blood Mini (Qiagen) or QIASymphony DSP DNA Mini Kit (Qiagen). The isolation process was followed by quality control and quantification using a Qubit 2.0 Fluorometer (Invitrogen) and a TapeStation 2200 system (Agilent). Libraries for whole-genome sequencing were prepared with

the Illumina TruSeq Nano (100 ng genomic DNA as input). Both tumor and control (germline) samples were sequenced on 2 lanes Illumina HiSeq X Ten (Supplementary Data 15). Libraries for whole-exome sequencing were prepared with the Agilent SureSelect All Exon Kit v5 or v5 + UTRs (200 ng input). The libraries were sequenced on Illumina HiSeq 2000, HiSeq 2500 or HiSeq 4000 (Supplementary Data 15). Samples were processed centrally by the NCT Molecular Precision Oncology Program Sample Processing Laboratory (SPL) and sequenced by the DKFZ Genomics and Proteomics Core Facility (GPCF). Further information and exceptions are listed in Supplementary Data 8.

*Nucleotide sequence alignment*. DNA sequencing reads were mapped to the assembly comprising human genome (1000 Genomes Phase 2 of the Genome Reference Consortium; version hs37d5) and a genome of Enterobacteria phage phiX174 using BWA mem (version 0.7.15) with -T0 parameter as the one different from the default. BAM files were sorted with bamsort (biobambam package, version 0.0.148), and duplicates were marked with markdup (Sambamba package, version 0.6.5)[48]. Sequencing quality statistics are summarized in Supplementary Data 15.

**Calling of single-nucleotide variants and small insertions and deletions**
*Somatic*. Somatic SNVs were detected from matched tumor/normal sample pairs by an in-house analysis pipeline based on SAMtools mpileup and bcftools and using heuristic filtering as previously described[49–51]. In short, initial SNV calls were detected in the tumor BAM by SAMtools (version 0.1.19) mpileup, which considered only reads with minimum mapping quality of 30 (-q 30), and BCFtools, which reported all positions containing at least one high-quality non-reference base (-vcgN -p 2.0). Afterwards these positions were checked in the control sample using mpileup. SNVs were then annotated with ANNOVAR (version November 2014) using GENCODE (release 19). Downstream filtering discarded variants with low support of the alternative allele, occurring in tandem repeats and other read-attracting regions, having PCR strand bias (WGS only), having sequencing strand bias, and having significant bias in the PV4 field of the mpileup output. SNVs with low confidence score were discarded. Somatic SNVs annotated as missense, stopgain, stoploss, or splicing (two base pairs next to an exon boundary) were defined as non-silent. Short indels were detected by Platypus (version 0.8.1) for matched tumor/normal sample pairs[52]. Only ones that had Platypus filter flag PASS or passed custom filters allowing for low variant frequency were retained. Annotation of short indels was done using ANNOVAR (version February 2016). The calls falling into a coding sequence or splice-site were extracted.

*Germline*. Germline indels were called by Platypus. SNVs identified in the tumor sample were annotated as germline if the control sample had at least 1/30 reads supporting the alternative allele. Germline variants in 101 cancer predisposition genes (Supplementary Data 4) were further filtered for rare variants and against frequent variants in an in-house database before assessment according to AMP-ACMG guidelines. The p-value for age at onset comparison was generated with a two-sided, equal variance t-test.

*Tumor ploidy, purity and copy number profile determination*. For samples sequenced with WGS, the absolute allele-specific copy numbers, tumor ploidy and purity were estimated using ACEseq (version 5.0.1)[53].

For samples sequenced with WES, the absolute allele-specific copy numbers were estimated using CNVkit (version 0.9.3)[54]. Segments containing at least 20 heterozygous SNPs were further processed to infer sample ploidy and tumor cell content (TCC) using a method adapted from ACEseq. The algorithm tested each possible combination of TCC (range 0.15-1.0) and ploidy (range 1.0–6.5) to find the local minima and thus optimal solution. If more than one solution was possible, they were visually evaluated and ultimately one of them was chosen. Samples with tumor cell content estimated to be 100% were considered unreliable (due to in fact low tumor cell content) and thus discarded from the results ($n = 19$).

*Microsatellite instability*. Microsatellite instability was detected with MSIsensor (version 0.2)[55]. The list of homopolymers and microsatellites generated with the MSIsensor scan command from the 1000 genomes reference comprises 33,386,244 loci. MSIsensor was run with a minimum required coverage of 15 reads for genomes and 30 for exomes in both tumor and control. A score > 3.5 implies microsatellite instability.

*RNA sequencing and gene fusion detection*. If RNA quality was sufficient, either the Illumina TruSeq RNA (with 1000 ng total RNA) or the Illumina TruSeq mRNA stranded protocol (with 500 ng total RNA) was used for library preparation (TruSeq mRNA stranded since February 2016, Supplementary Data 15). Both are Oligo-dT-based protocols and enrich for mRNA only. Three libraries were pooled and sequenced on one lane HiSeq 4000 100 PE. The reads were aligned to the same reference genome as DNA sequencing data with STAR 2.5.1b[56]. The gene fusions were detected by Arriba pipeline (version 0.8), the software is available on GitHub[57]. Fusions were categorized into high, medium or low level of confidence.

*Mutational signatures.* Mutational signatures were calculated using R/Bioconductor package YAPSA (version 1.13.3)[58] and COSMIC signatures (version 2)[59]. All identified somatic SNVs were used for the analysis. Six samples with less than 50 SNVs were excluded from the analysis. Mutational catalogs were calculated separately for the whole-exome and the whole-genome sequencing data. The whole-exome catalog was corrected additionally by factors specific for the target capture kits that were used for the preparation of samples. Afterwards, mutational catalogs were normalized by the average length of the coding sequence in Mb (2800 and 30 for WGS and WES, respectively) and merged together for signature decomposition. Exposures were calculated per sample using the set of 30 validated signatures (no artifact signatures) and absolute signature-specific cutoffs with cost factor 6. Corresponding confidence intervals were calculated per sample. Only if their lower bound was greater than 0, the signature was considered to be positively identified.

*Methylation-based entity prediction.* The Infinium MethylationEPIC BeadChip microarray (850 K) was used for 55 CUP samples and 77 samples of the validation cohort (Supplementary Data 7) to interrogate DNA methylation patterns at genome-wide level. All samples were gathered within the NCT/DKTK MASTER program and had a tumor cell content >30% of the NCT/DKTK MASTER cohort to interrogate DNA methylation patterns at the genome-wide level. The library preparation, hybridization and scanning of the array was performed at the German Cancer Research Center (DKFZ) Genomics & Proteomics Core Facility. The raw data (idat files) were processed into beta values with the minfi R package[60]. Beta values range from 0 to 1 with 0 being a CpG unmethylated and 1 fully methylated.

TCGA (The Cancer Genome Atlas) pan-cancer methylation was retrieved via the curatedTCGAData package[61]. The dataset (33 entities, 8024 samples) comprised both 450k and 27k methylation arrays. The intersection of these arrays comprised 25978 CpGs. For a more meaningful entity prediction colorectal (COAD) and rectal (READ) adenocarcinomas were binned together (COAD/READ).

The 5000 CpGs for the methylation-based entity predictions were derived after (i) removing known SNPs as previously described[62], (ii) only considering overlapping CpGs between 850k, 450k and 27k arrays to ensure compatibility with all Illumina methylation arrays and (iii) lastly calculating the top 5000 most variant CpGs across the pan-cancer dataset. The probe IDs are listed in Supplementary Data 16.

Methylation-based entity prediction of 55 CUP samples and 77 samples of the validation cohort was performed by correlating (Spearman correlation) the vector of 5000 CpGs with all samples in the TCGA cohort. The entity of the sample with the highest correlation coefficient was deemed to be the predicted entity.

Similarity in methylation profiles was visualized by tSNE plot with the Rtsne R package[63]. Missing data was imputed with the impute.knn function. The perplexity was set to 100.

*Transcriptome-based entity prediction.* In order to identify the tissue of origin of a CUP sample based on gene expression, we searched for samples with a similar expression profile in two reference cohorts: the MASTER cohort (comprising 1890 samples from 1814 patients, Supplementary Data 9) and the union of 33 TCGA cohorts (TCGA cohorts with >50 samples were subsampled, yielding a total of 1809 samples). For each reference cohort, we compared the expression profile of the CUP sample to all possible pairwise combinations of reference samples. The reference samples were ranked by the number of times they were more similar to the CUP sample than the other reference sample in a given pair of reference samples. Similarity was measured as the fraction of genes that were upregulated in both the CUP sample and one of the samples in a given pair of reference samples (FPKM > 13), but downregulated in the other reference sample (FPKM < 3). The thresholds for up- and downregulation were determined by means of 10-fold cross-validation on a subset of the MASTER cohort. To mitigate the distortion of the CUP expression profile by contamination from surrounding normal tissue in the bulk RNA-Seq data, we ignored genes found to be upregulated in normal liver tissue (Supplementary Data 17) if the sample was obtained by liver biopsy. The entity of the most similar reference sample was assumed to predict the entity of the CUP sample. If the most similar reference sample was a CUP as well, the most similar non-CUP sample was chosen for prediction instead. The method was validated on 72 patients from the validation cohort (Supplementary Data 6).

*Tumor mutational burden (mutations per megabase).* For each sample, the numbers of non-silent SNVs and coding indels in the exons of the tumor were added and divided by the length of the coding sequence of the genome (in Mb). The denominator depended on the technology, including different library preparation kits, used for sequencing of a sample. For samples sequenced with WGS, the GENCODE Human v19 gene annotation (GTF format) was taken, coding sequences were identified and merged, and the total length was calculated. For samples sequenced with WES, however, the merged coding sequences were additionally intersected with the coordinates of the corresponding target capture (BED format). All sequence operations were done using bedtools v2.27.1[64]. Calculations resulted in lengths: (i) 35.334619 Mb for WGS, (ii) 31.057260 Mb for WES with

SureSelectXT Human All Exon V5 including UTRs and (iii) 30.894643 Mb for WES with SureSelectXT Human All Exon V5 excluding UTRs. Please note that the MTB used the sum of non-silent SNVs and coding indels as measure for TMB.

*Homologous recombination deficiency.* Homologous recombination deficiency (HRD) was determined using three different methods. The first one was being used for Molecular Tumor Board and could be applied to both whole-exome and whole-genome sequencing data. This method was based solely on results from the copy number analysis, and consisted of the estimation of two parameters: loss of heterozygosity (LOH-HRD)[65] and large-scale state transitions (LST)[66]. An unweighted sum of those produced a score, which classified samples to high (>20), intermediate (11–20) or low (≤10) level of impaired homologous recombination.

The other two methods which were used, HRDetect[67] and CHORD (version 2.0)[68], calculate a method-specific probability score of HR deficiency and can be applied to whole-genome sequencing data only. As inputs, they both used raw data comprising single-nucleotide variants, small insertions and deletions, structural variants (detected with SOPHIA, https://bitbucket.org/utoprak/sophia/src/master) and, only in case of HRDetect, copy number variation. All 27 whole-genome sequencing samples with reliable copy number data were therefore used in the analysis. Genes considered as HRD related in Fig. 1 are listed in Supplementary Data 18.

*Viral infections.* We used three computational approaches to detect viral infections from next-generation sequencing (NGS) data: a *k*-mer-based approach (Kraken2 version 2.1.2[69]), an assembly-based approach (P-DiP[70]), and an alignment-based approach where the sequencing reads were aligned against concatenated assemblies of the human genome and all RefSeq viral genomes, in accordance with Arriba's workflow for the detection of viruses. Kraken2 was considered to make a call if it detected at least one read per 40 million mapped reads as originating from a virus and if at least 10% of the viral genome was covered with reads. For P-DiP, a cutoff of one virus-originating read per million mapped reads was used. Moreover for the Arriba workflow, a sample was considered to be associated with a virus when at least 5% and 100 bp (whichever was bigger) of the viral genome was covered with reads. Supplementary Table 1 lists all viruses that were reported by at least two methods. To detect viral integration sites, we used Arriba version 2.1.0 for RNA-Seq data and VIRUSBreakend version 2.12.0[71] for DNA-Seq data.

*Additional data processing and analysis.* The downstream analysis was performed in R (version 3.4.3) using Bioconductor repository and such packages as tidyverse (version 1.2.1)[72], ComplexHeatmap (version 1.99.5)[73] and Biobase (version 2.38.0)[74]. If possible, the sample used for the first MTB was used for the general cohort description, only for CUP-70 we analyzed the sample for the second MTB. PFS, PFSr, and mPFSr were calculated using Microsoft Excel. Survival analysis using Kaplan–Meier estimator and log-rank tests was performed using ggplot2 (version 3.3.3). *p*-values < 0.05 were considered statistically significant.

**Reporting summary.** Further information on research design is available in the Nature Research Reporting Summary linked to this article.

## Data availability
TCGA pan-cancer methylation was retrieved via the curatedTCGAData package[61]. FPKM expression values of the TCGA cohorts were obtained from the GDC data release v22.0. Genome, transcriptome and methylation data generated in this study have been deposited in the European Genome-phenome Archive under the accession number EGAS00001004786. The data are available under controlled access due to the sensitive nature of genome sequencing data, and access can be obtained by contacting the appropriate Data Access Committee listed for each dataset in the study. Access will be granted to commercial and non-commercial parties according to patient consent forms and data transfer agreements for as long as needed. We have an institutional process in place to deal with requests for data transfer and aim for rapid response time. GENCODE (release 19) was used for gene annotation and is publicly available. The raw clinical data are protected and are not available due to data privacy laws. The processed clinical data are available as Supplementary Data files. The remaining data are available within the Article, Supplementary Information, Supplementary Data or Source Data file. Source data are provided with this paper.

## Code availability
Bioinformatics analyses were performed using above-mentioned open-source software with parameters as described in each method section. The R script for transcriptome-based entity prediction is available in Supplementary Software 1.

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

## Acknowledgements

We thank the NCT Molecular Precision Oncology Program Sample Processing Laboratory and the DKFZ Genomics and Proteomics Core Facility for technical support. We also thank Viktoria Brendel, Jana-Viktoria Maier, and Peter Lichter for infrastructure and program development within the NCT Molecular Precision Oncology Program. This work was supported by the NCT Molecular Precision Oncology Program, grant H021 from the DKFZ-Heidelberg Center for Personalized Oncology, and the DKTK Joint Funding Program. Maximilian Werner was supported by scholarships of the German Academic Scholarship Foundation (Studienstiftung des deutschen Volkes) and the Heinrich-Boell-Foundation. Lino Möhrmann was partly supported by the Else Kröner Research College Dresden (Clinician Scientist Program led by Professor A. El-Armouche). Rainer Hamacher was supported by the Clinician Scientist Program of the University Medicine Essen Clinician Scientist Academy (UMEA) sponsored by faculty of medicine and Deutsche Forschungsgemeinschaft (DFG). Anna L. Illert was supported by the Mildred-Scheel-Professorship Program of the German Cancer Aid (#70114112).

## Author contributions

Conception and design: L.M., H.G., and S.F. Data acquisition and assembly: L.M., M.W., M.O., A.M., A.J., S.U., S.K., M.F., B.H., C.E.H., V.T., D.B.L., M.Z., D. Hanf, C.L., M.A., R.P., G.R., I.J., R.H., J.F., S.W., C.H.B., M.B., A.L.I., K.H.M., C.B.W., A.D., T.K., G.F., W.W., B.B., A.S., E.S., D. Hübschmann, P.H., C.H., S.F., and H.G. Bioinformatic analysis: M.O., A.M., B.H., M.F., S.U., N.P., B.B., and D. Hübschmann. Project, sample, and data management: D.R., K.B., U.W., and K.P. Clinical data curation and analysis: L.M., M.W., A.J., and C.H. Data analysis and interpretation: L.M., M.W., M.O., S.U., A.M., A.J., S.U., C.H., S.F., and H.G. Manuscript writing: L.M., M.W., M.O., C.H., S.F., and H.G. Manuscript revision, editing, and approval: All authors.

## Funding

## Competing interests

A.D.: received personal and speakers' fees, reimbursement for travel and accommodation and honoraria for participation in advisory boards from Astra-Zeneca, Bayer, BMS, Eisei, MSD, Pfizer, Janssen-Cilag, Roche & Servier. A.S.: provided consultation, attended advisory boards, or provided lectures for Astra Zeneca, AGCT, Bayer, Bristol-Myers Squibb, Chugai, Eli Lilly, Illumina, Janssen, Merck Sharp & Dohme, Novartis, Pfizer, F Hoffmann-La Roche, Seattle Genetics, Takeda and Thermo Fisher. A.J.: Honoraria: AstraZeneca. A.L.I.: has attended Advisory Boards and served as speaker for Roche, AstraZeneca, Ars Tempi, Takeda, Janssen-Cilag, AbbVie and Incyte outside of the submitted work. C.B.W.: received personal and speakers' fees, reimbursement for travel and accommodation and honoraria for participation in advisory boards from Bayer, BMS, Celgene, GSK, Ipsen, MedScape, Merck, Rafael Pharmaceuticals, RedHill, Roche, Servier, Shire/Baxalta, SirTex, and Taiho and scientific grant support by Roche. C.H.: Honoraria: Roche, Novartis; research funding: Boehringer Ingelheim; advisory board: Boehringer Ingelheim. J.F.: Honoraria: Pharmamar, Decidphera; Travel Support: Pharmamar, Eli Lilly outside of the submitted work. L.M.: non-financial support (travel support) from Celgene outside of the submitted work. M.F.: her husband works for Merck KGaA. R.H.: Travel Support: Lilly, Novartis and PharmaMar as well as fees from Lilly and PharmaMar outside of the submitted work. W.W.: has attended Advisory Boards and served as speaker for Roche, MSD, BMS, AstraZeneca, Pfizer, Merck, Lilly, Boehringer, Novartis, Takeda, Bayer, Amgen, Astellas, Illumina, Molecular Health, NewOncology and Agilent. W.W. receives research funding from Roche, MSD, BMS and AstraZeneca. S.F.: Consulting or advisory board membership: Bayer, Illumina, Roche; honoraria: Amgen, Eli Lilly, PharmaMar, Roche; research funding: AstraZeneca, Pfizer, PharmaMar, Roche; travel or accommodation expenses: Amgen, Eli Lilly, Illumina, PharmaMar, Roche. The other authors declare no potential conflicts of interest.

## Additional information

¹Department of Translational Medical Oncology, National Center for Tumor Diseases (NCT) Dresden and German Cancer Research Center (DKFZ), Heidelberg, Germany. ²Center for Personalized Oncology, NCT Dresden and University Hospital Carl Gustav Carus, Faculty of Medicine and Technische Universität Dresden, Dresden, Germany. ³German Cancer Consortium (DKTK), Dresden, Germany. ⁴Computational Oncology Group, NCT Heidelberg and DKFZ, Heidelberg, Germany. ⁵Department of Translational Medical Oncology, NCT Heidelberg and DKFZ, Heidelberg, Germany. ⁶Department of Medical Oncology, NCT Heidelberg and Heidelberg University Hospital, Heidelberg, Germany. ⁷Molecular Precision Oncology Program, NCT Heidelberg, Heidelberg, Germany. ⁸Institute for Clinical Genetics, University Hospital Carl Gustav Carus Dresden, Technische Universität Dresden, Dresden, Germany. ⁹Section Translational Cancer Epigenomics, Department of Translational Medical Oncology, NCT Heidelberg and DKFZ, Heidelberg, Germany. ¹⁰Division of Molecular Genetics, DKFZ, Heidelberg, Germany. ¹¹Institute of Pathology, University Hospital Heidelberg, Heidelberg, Germany. ¹²Charité Comprehensive Cancer Center, Universitätsmedizin Berlin, Freie Universität Berlin and Humboldt-Universität zu Berlin, Berlin, Germany. ¹³Department of

Medical Oncology, West German Cancer Center, University Hospital Essen, Essen, Germany. [14]DKTK, Essen, Germany. [15]Department of Medicine 2, Hematology/Oncology, Goethe University, Frankfurt, Germany. [16]DKTK, Frankfurt, Germany. [17]Institute of Medical Bioinformatics and Systems Medicine, Medical Center—University of Freiburg, Faculty of Medicine, University of Freiburg, Freiburg im Breisgau, Germany. [18]DKTK, Freiburg, Germany. [19]Department of Internal Medicine I, Medical Center—University of Freiburg, Faculty of Medicine, University of Freiburg, Freiburg, Germany. [20]Department of Internal Medicine III, University Hospital, LMU Munich and Comprehensive Cancer Center, Munich, Germany. [21]Department of Hematology and Cellular Therapy, University Hospital Leipzig, Leipzig, Germany. [22]University Cancer Center Mainz, University Medical Center Mainz, Mainz, Germany. [23]DKTK, Mainz, Germany. [24]Institute of Pathology, Technical University Munich, Munich, Germany. [25]Division of Applied Bioinformatics, NCT Heidelberg and DKFZ, Heidelberg, Germany. [26]Pattern Recognition and Digital Medicine, Heidelberg Institute for Stem cell Technology and Experimental Medicine, Heidelberg, Germany. [27]DKTK, Heidelberg, Germany. [28]Translational Functional Cancer Genomics, NCT Heidelberg and DKFZ, Heidelberg, Germany. [29]These authors contributed equally: Lino Möhrmann, Maximilian Werner, Małgorzata Oleś, Andreas Mock, Sebastian Uhrig. [30]These authors jointly supervised this work: Christoph Heining, Stefan Fröhling, Hanno Glimm. ✉email: hanno.glimm@nct-dresden.de

