## [Peer Review File · Nature Communications]

Comprehensive genomic and epigenomic analysis in cancer of unknown primary guides molecularly-informed therapies despite heterogeneityReviewers' Comments:

Reviewer #1:

Remarks to the Author:

Möhrmann and colleagues have described a multi-omics analysis of cancers of unknown primary. The study describes a) frequencies of somatic and pathogenic germline gene mutations across the cohort b) comparison of RNA and DNA methylation based tissue of origin classifiers c) details of some potential diagnostic mutational features and d) description of clinical outcome in a subset of patients given genomics directed treatments. The cohort size is modest (n=70) compared to other previously published series using panel DNA sequencing (e.g. JAMA Oncol 2015 Vol. 1 Issue 1 Pages 40-9 (n=200) , Ann Oncol 2017 Vol. 28 Issue 12 Pages 3015-3021 (n=150)). The reported frequency of specific gene mutations is similar to that reported in prior studies and few conclusions can really be made from the data other than CUP is a molecularly heterogeneous disease, which is already well known. It may also be questionable if the series is truly representative of CUP given the relatively young average age of the patients and inclusion of non-carcinoma types including melanoma and small round cell tumors. Contrasting results from two orthogonal molecular classifiers is one very interesting aspect of the study but was quite poorly executed. Similarly, the interpretation and reporting of mutational signatures from WGS and WES data could have been done better and in this regard the study fails to harness one of the major benefits of taking a genome-wide and multi-omics approach. In summary, I felt the study could have been done better. As it stands it does not really provide any major insights into our understanding of CUP biology or highlight the application multi-omics analysis of CUPs using current "best practice" methods. As a clinical report I think it is still an important study highlighting the importance of molecular profiling for clinical decision-making in CUP.

Comments

1. Mutational signature analysis. The description and interpretation of COSMIC mutational signatures needs refinement. For example, a UV signature is described in 12 cases which may imply a potential skin origin- an unusually high fraction of such cases among CUP tumors. A UV signature is usually detected in the context of high mutational burden but this does not appear to be the case in the majority of CUPs given only three tumors in the whole series have TMB >10mut/Mb. Was the UV signature dominant in all cases where it was detected? Similarly, smoking sig4 is described in 9 case in the manuscript but is only noted as a useful feature in 3 cases in Supp Table 5. If the authors are confident there is a dominant mutational signature such as UV (sig 7) or smoking (sig 4) then these features have important diagnostic utility and this information should be presented in an interpretable way.
2. HRD prediction. With regards to HRD the authors reported LOH and LST patterns with combined combined scores as well as sig3. Were both sig 3 and LOH/LST features used together to interpret HRD for directing PARPi? In Supp Table 6.1 it appears that those cases with reported sig 3 had low or medium LOH/LST scores (e.g. CUPs 54 and 66) yet these cases also had BRCA2 mutations. More reliable methods for predicting HRD in pan-cancer data have been developed using WGS and these methods are superior to genomic scar-based indexes or sig3 alone. These newer methods include HRDetect (Nat Med 2017 Vol. 23 Issue 4 Pages 517-525) and CHORD (Nat Commun 2020 11, 5584. CHORD is freely available (<https://github.com/UMCUGenetics/CHORD>))
3. Tissue of origin classification. A major deficiency here was that the performance of the classifiers was not assessed using an independent set of metastatic tumors of known origin. This is an essential step in developing and validating a ToO classifier otherwise overfitting of the data and/or technical biases between training and test sets cannot be ruled out. Classifying a tumor based on the nearest paired tumor sample in the reference by Spearman correlation really lacks the sophistication of methods described in other similar published CUP classifier studies. Multi-class classification and machine learning methods are readily available including k-nearest neighbor, support vector machines or neural network. Typically, these classifiers are tuned using leave-one-out cross validation on the training data and before testing on independent test set. Without this detail included in the manuscript the performance of the classifier cannot be trusted and neither can the results reported on CUP.

4. Supporting histopathology data. It is stated that ESMO guidelines were used for selecting CUP cases but very little detail is provided about any immunohistochemistry staining done. A CK7 and CK20 staining profile should be presented if this data is available in addition to any other Dx stains used. Melanoma of unknown primary should be excluded from the CUP series.
5. Virus DNA detection. This is an important molecular feature that can be detected by WGS and RNA-seq but has not been included.
6. Figure 1 should highlight platform used (WGS/WES).
7. Figure 2. How does tSNE clustering relate to the two ToO classifier methods employed? How is the reader to interpret this data?
8. Fusions. Anecdotal reports of fusions in CUPs are interesting but specific fusions that reported anecdotally in other cancer types cannot be used as pathognomonic features. e.g. EZR-ERBB4 has only been reported in only one mucinous lung cancer therefore cannot be used as evidence to support a lung origin.
9. There appears to be some inconsistencies in the data relating to TMB. For instance, CUPs 16, 18, 21 and 51 are reported as hypermutated in Supp Table 6.1 based on > 100mut detected but in Figure 1 all cases have <10mut/Mb. How is the reader to compare and interpret these two TMB metrics? Cases with <10mut/Mb would not seem hypermutated.

Reviewer #2:

Remarks to the Author:

Möhrmann and colleagues report on comprehensive and integrative analyses on genomic, transcriptomic and methylomic level from a subpopulation of patients with CUP syndrome enrolled into the MASTER program. The analysis population comprises 70 patients.

Patients with CUP syndrome have a poor prognosis; several attempts have been made to improve outcome e.g. by identifying site of origin. Besides non-comparative studies, one randomized trial has been reported aiming to better classify CUP patients regarding tumor site of origin and recommend respective treatment, however, as also discussed by the authors, this trial could not show that this approach actually leads to improved patient survival. Another randomized trial (SHIVA trial) compared treatment and outcome of patients with exhausted treatments options (not CUP patients) who received either standard of care or treatment as recommended by a molecular tumor board. The SHIVA trial was reported in 2016 and, unfortunately, could not show a substantial benefit for this approach either.

In brief, the manuscript adds to the existing body of evidence of translational diagnostics and treatment approaches for CUP syndrome or patients with exhausted treatment options. The overall number of patients is smaller compared to similar studies in the field and the patient population is likely not representative for CUP patients, because of the obvious bias towards younger patients. It provides some new aspects, and it reveals that such wealth and in-depth analyses can be contradictory and that there still remain many open questions who to translate such findings into patient care.

In the current version of the manuscript, there are several open issues that need to be addressed. This study is primarily a clinical study; thus, it should clearly report items as strongly recommended in reporting guidelines such as STROBE or REMARK. Especially regarding definition of analysis populations, calculation and reporting of clinical outcome parameters. I have the following specific comments that may help to improve clarity of the paper:

1. Where there any general specific inclusion/exclusion criteria for the MASTER program? If yes, please report these criteria or provide them in the supplement. You refer to references 30 and 43, but this is not sufficient for the reader.
2. Is the MASTER program registered with a clinical trial registry? If yes, please provide the respective

registration identifier. If it was not registered, please provide the reason.

3. Did you collect clinical patient data prospectively? You somehow describe this in section "Clinical and statistical analysis", but this is very superficial. How have clinical data been captured? Was there a centrally managed EDC system? Please comment.

4. What type of sample (core needle biopsy, resection of metastases) was required for participation in the MASTER program? Were analyses based on fresh biopsy at progression and / or at primary diagnosis? Please report on what types of samples you conducted the analyses.

5. Please provide a statement on the type of design of your study. Given the information provided, I anticipate this is an observational prospective cohort study, but this needs clear statement in the paper.

6. I cannot find any section commenting on written informed consent of patients and ethical review of the study. Especially regarding the germline analysis, patients usually have to provide additional written consent. Please add this to the manuscript.

7. You obviously used the program R for the bioinformatical part, but I cannot find a description of the statistical program you used to calculate the clinical outcome parameters such as PFS and response. Please comment on this and provide description in the methods section.

8. It is not clear to me whether reported patients were included from one or several centers. You mention the DKTK consortium, but this is not really clear for readers not being familiar with current structures in Germany. Was this a single or multicenter study? If multicenter, from how many centers were patients included?

9. Where have the analyses been conducted? Was it one laboratory or several sites? If yes, how was standardization of assays etc. guaranteed?

10. Did you pre specify tests and panels to be done? E.g. a minimal core set of WES followed by other techniques / analyses if sufficient tissue was available?

11. Regarding the entity prediction, were results from these analyses available to the MTB when discussing the cases and making recommendations?

12. The reported 70 patients were relatively young and are likely not representative for patients usually diagnosed and treated with CUP syndrome. How do you explain this? May this be because of your inclusion/exclusion criteria?

13. You report to have identified and included 70 patients with CUP syndrome, however, only 61 patients fulfilled the ESMO criteria. I assume this is caused by discrepancy between inclusion criteria of the MASTER program and the ESMO criteria. Why is this? Please comment on this.

14. Please provide percentage in addition to the absolute numbers in table 1.

15. One of most central major revisions required is the definition of analysis populations and outcome reporting. Current presentation of clinical outcome is certainly biased.

16. Specifically: First, all 70 patients should be in the denominator when reporting outcome, thus, for all 70 patients PFS and OS should be calculated and reported. Also, for all these 70 patients you should calculate the PFS ratio irrespective of the MTB recommendation or whether recommended treatment has been given. Second, if you want to explore potential associations of benefit between personalized treatment and outcome, you need at least to contrast outcome (PFS, response, PFS ratio,

OS) of patients who received the recommended treatment (N=20), but also the outcome of those patients who had a recommendation but did not receive the recommended treatment (N=36). A stratified table, similar to content of table 1, would be helpful allowing to compare patient and disease characteristics. In addition, information on the evidence level of the MTB recommendation would also enrich the content. However, such exploratory analysis would still not allow to conclude that personalized treatment based on an MTB leads to improved patient outcome, because of several unmeasured confounding factors. For such statement you need a randomized trial. This limitation also needs to be addressed in the discussion.

17. For all Kaplan-Meier plots, please provide 95% confidence intervals and at-risk table below the graph as this is standard in reporting time-to-event data.

18. Please report the absolute number of PFS defining events together with PFS. Please also report OS and the absolute number of deaths. Finally, also report the follow-up time of all patients, e.g. using the inverse Kaplan-Meier method.

19. How was treatment in accordance with the MTB recommendation defined? Please specify this.

20. Where there any pre-specified assessments or where patient outcomes determined based on routinely collected data? Please specify.

21. How was PFS calculated? Please provide the starting date for calculating the PFS as well as definition of PFS defining events.

22. How did you collect information PFS of the most previous therapy before inclusion into the MASTER program?

23. Have you collected the reasons why recommended treatments were not applied? Was this a reimbursement issue? This is a very central issue in all initiatives of personalized oncology treatment. Please comment.

24. Please report all outcomes in months, not in days.

NCOMMS-21-08986A

> We thank the reviewers for their valuable comments. We have addressed comments as requested which has profoundly improved the quality of our manuscript. Below you will find our answers to each comment.

Reviewer 1

Reviewer #1, expert in genomics for precision oncology (Remarks to the Author):

Möhrmann and colleagues have described a multi-omics analysis of cancers of unknown primary. The study describes a) frequencies of somatic and pathogenic germline gene mutations across the cohort b) comparison of RNA and DNA methylation based tissue of origin classifiers c) details of some potential diagnostic mutational features and d) description of clinical outcome in a subset of patients given genomics directed treatments. The cohort size is modest (n=70) compared to other previously published series using panel DNA sequencing (e.g. JAMA Oncol 2015 Vol. 1 Issue 1 Pages 40-9 (n=200) , Ann Oncol 2017 Vol. 28 Issue 12 Pages 3015-3021 (n=150). The reported frequency of specific gene mutations is similar to that reported in prior studies and few conclusions can really be made from the data other than CUP is a molecularly heterogeneous disease, which is already well known. It may also be questionable if the series is truly representative of CUP given the relatively young average age of the patients and inclusion of non-carcinoma types including melanoma and small round cell tumors. Contrasting results from two orthogonal molecular classifiers is one very interesting aspect of the study but was quite poorly executed. Similarly, the interpretation and reporting of mutational signatures from WGS and WES data could have been done better and in this regard the study fails to harness one of the major benefits of taking a genome-wide and multi-omics approach. In summary, I felt the study could have been done better. As it stands it does not really provide any major insights into our understanding of CUP biology or highlight the application multi-omics analysis of CUPs using current "best practice" methods. As a clinical report I think it is still an important study highlighting the importance of molecular profiling for clinical decision-making in CUP.

1. Mutational signature analysis. The description and interpretation of COSMIC mutational signatures needs refinement. For example, a UV signature is described in 12 cases which may imply a potential skin origin- an unusually high fraction of such cases among CUP tumors. A UV signature is usually detected in the context of high mutational burden but this does not appear to be the case in the majority of CUPs given only three tumors in the whole series have TMB >10mut/Mb. Was the UV signature dominant in all cases where it was detected? Similarly, smoking sig4 is described in 9 case in the manuscript but is only noted as a useful feature in 3 cases in Supp Table 5. If the authors are confident there is a dominant mutational signature such as UV (sig 7) or smoking (sig 4) then these features have important diagnostic utility and this information should be presented in an interpretable way.

> We have extended the presentation of results on mutational signature analysis for signatures with diagnostic potential, namely AC3, AC4, AC7 and AC11. First, we included the number of patients per dominant signature in the main text. Secondly, we added figures S3 b-e from which the reader can not only see the dominance level of each relevant signature in samples in which it was detected, but also both non-normalized and normalized exposure values. Where applicable, information on dominant signatures was added to the results for tissue origin prediction included in Supplementary Table 6.

Nonetheless, it needs to be considered that mutational signatures corresponding to ultraviolet light exposure and tobacco smoking have been previously described in cancers of origin outside skin and lung, respectively (AC7:

https://cancer.sanger.ac.uk/signatures/media/images/v3.2_SBS7a_TISSUE.original.jpg,

https://cancer.sanger.ac.uk/signatures/media/images/v3.2_SBS7b_TISSUE.original.jpg,

https://cancer.sanger.ac.uk/signatures/media/images/v3.2_SBS7c_TISSUE.original.jpg,

https://cancer.sanger.ac.uk/signatures/media/images/v3.2_SBS7d_TISSUE.original.jpg,

AC4: https://cancer.sanger.ac.uk/signatures/media/images/v3.2_SBS4_TISSUE.original.jpg).

This data has been published by Alexandrov et al. in Nature, 2020 (<https://doi.org/10.1038/s41586-020-1943-3>).

Moreover, based on that data, both signatures have been frequently identified in samples with TMB < 10mut/Mb. Therefore, we are careful with the interpretation of mutational signature analysis results for the diagnostic purposes in CUPs. However, we greatly appreciate the reviewers' suggestion to focus on signature dominance level. For example, in samples with dominant signature AC7, tissue of origin was unanimously predicted by all tools tested by us to be skin. Signature dominance is a valid and important clue about the tissue of origin, but in order to be a strong one, we believe, it still should be accompanied by more evidence. We have also updated the corresponding discussion part.

2. HRD prediction. With regards to HRD the authors reported LOH and LST patterns with combined combined scores as well as sig3. Were both sig 3 and LOH/LST features used together to interpret HRD for directing PARPi? In Supp Table 6.1 it appears that those cases with reported sig 3 had low or medium LOH/LST scores (e.g. CUPs 54 and 66) yet these cases also had BRCA2 mutations. More reliable methods for predicting HRD in pan-cancer data have been developed using WGS and these methods are superior to genomic scar-based indexes or sig3 alone. These newer methods include HRDetect (Nat Med 2017 Vol. 23 Issue 4 Pages 517-525) and CHORD (Nat Commun 2020 11, 5584. CHORD is freely available (<https://github.com/UMCUGenetics/CHORD>))

> We thank the reviewer for pointing out these additional methods that have been published recently. We have added both of them, HRDetect and CHORD. They have enriched our analysis and allowed us to correlate their results with our method that is being used for the molecular tumor board in MASTER. Both HRDetect and CHORD predict HRD in WGS samples, whereas more than half of our cohort data comes from WES. Results of both methods are in line with our method. When discussing patients in the molecular tumor board we depend on methods that can be applied broadly to as many patients as possible. Therefore, we have reported our method as it was used in the molecular tumor board. HRD was interpreted based on a summary of signature AC3, LOH/LST scores, somatic and germline mutations and used as an argument for recommending PARPi.

3. Tissue of origin classification. A major deficiency here was that the performance of the classifiers was not assessed using an independent set of metastatic tumors of known origin. This is an essential step in developing and validating a ToO classifier otherwise overfitting of the data and/or technical biases between training and test sets cannot be ruled out. Classifying a tumor based on the nearest paired tumor sample in the reference by Spearman correlation really lacks the sophistication of methods described in other similar published CUP classifier studies. Multi-class classification and machine learning methods are readily available including k-nearest neighbor, support vector machines or neural network. Typically, these classifiers are tuned using leave-one-out cross validation on the training data and before testing on independent test set. Without this detail included in the manuscript the performance of the classifier cannot be trusted and neither can the results reported on CUP.

> We agree with the reviewer, which is why we have addressed this issue by setting up a validation cohort using 100 consecutive MASTER patients enrolled between 12/2020 and 06/2021. We only included entities that are part of TCGA to ensure comparability with other tissue-of-origin classifiers, which were usually trained on TCGA data and can therefore only reliably predict entities that are part of TCGA. We compared our results with two other published methods, cancerSCOPE (49 – 56 % accuracy) and CUP-AI-Dx (54% accuracy), both of which had similar accuracy as our entity prediction when using TCGA as a reference cohort (53% accuracy), but poorer accuracy than our entity prediction with MASTER as a reference cohort (75%). We have rewritten the respective result part of the manuscript and added the detailed validation cohort as supplementary information.

Furthermore, Zheng and colleagues published a DNA methylation-based deep neural network model with a very convincing accuracy (<https://doi.org/10.1371/journal.pone.0226461>). However, code documentation is insufficient to apply the trained algorithm to our MASTER cohort methylation data set.

4. Supporting histopathology data. It is stated that ESMO guidelines were used for selecting CUP cases but very little detail is provided about any immunohistochemistry staining done. A CK7 and CK20 staining profile should be presented if this data is available in addition to any other Dx stains used. Melanoma of unknown primary should be excluded from the CUP series.

> Available immunohistochemistry data was added as supplementary information in Supplementary Table 1.3. However, the diagnostic workup for CUP according to ESMO guidelines includes melanoma. Melanoma of unknown primary (MUP) is an entity that can and should be distinguished from melanoma with known primary. Therefore, we have not excluded MUP patients from our cohort since we took into consideration all cancers of unknown primary and not just carcinomas.

5. Virus DNA detection. This is an important molecular feature that can be detected by WGS and RNA-seq but has not been included.

> We agree with the reviewer and therefore, we have performed several virus detection methods based on DNA and RNA. The results and a corresponding method section have been added to the paper and its supplementary information. We used three computational approaches to detect viral infections from next-generation sequencing (NGS) data: a k-mer-based approach (Kraken2 version 2.1.2 [<https://doi.org/10.1186/s13059-019-1891-0>]), an assembly-based approach (P-DiP [<https://doi.org/10.1038/s41588-019-0558-9>]), and an alignment-based approach where the sequencing reads were aligned against concatenated assemblies of the human genome and all RefSeq viral genomes, in accordance with Arriba's workflow for the detection of viruses. To detect viral integration sites, we used Arriba version 2.1.0 for RNA-Seq data and VIRUSBreakend version 2.12.0 [<https://doi.org/10.1093/bioinformatics/btab343>] for DNA-Seq data. Three tumors were positive for human papillomavirus type 16 and another one was positive for human papillomavirus type 18. Integration of viral DNA into the host genome was detected in three cases inside/near the genes HOXA2, MIER1, SLC35D1, and TENM4. The respective result and method parts were updated accordingly.

6. Figure 1 should highlight platform used (WGS/WES).

> Figure 1 has been updated accordingly.

7. Figure 2. How does tSNE clustering relate to the two ToO classifier methods employed? How is the reader to interpret this data?

> We apologize that our figure legend did not explain the figure properly. We have updated the figure legend in order to clarify that Figure 2 A and B are a visualization of our methylation based

predictions. While many TCGA entities show very distinctive clusters, some do not. The updated figure legend helps to interpret this data properly.

8. Fusions. Anecdotal reports of fusions in CUPs are interesting but specific fusions that reported anecdotally in other cancer types cannot be used as pathognomonic features. e.g. EZR-ERBB4 has only been reported in only one mucinous lung cancer therefore cannot be used as evidence to support a lung origin.

> We agree that a single molecular alteration is usually not sufficient to suggest a certain underlying entity. This is in line with our statement in the result section “Diagnostic interpretation of molecular profiling” in which we have stated “Not all of them correlated with entity prediction results based on transcriptome profiling and these alterations alone did not necessarily justify diagnostic reclassification. Still, together with omics-based entity predictions they offer meaningful information of diagnostic value and can be useful for further treatment decisions.” However, we have updated the result text to state clearly that the EZR-ERBB4 fusion has been reported in only one mucinous lung cancer case.

9. There appears to be some inconsistencies in the data relating to TMB. For instance, CUPs 16, 18, 21 and 51 are reported as hypermutated in Supp Table 6.1 based on > 100muts detected but in Figure 1 all cases have <10mut/Mb. How is the reader to compare and interpret these two TMB metrics? Cases with <10mut/Mb would not seem hypermutated.

> We apologize for not explaining this properly. There are different ways of measuring TMB (Table 1, <https://doi.org/10.1002/gcc.22733>). Historically, MASTER has used 100 SNVs + indels as cut-off for defining hypermutation. All patients in this cohort received molecular tumor board recommendations using this cut-off. Therefore, this is what we reported in the respective result part. However, over the last couple of years measurement using mutations/Mb has become widely used. Therefore, we used mutations/Mb for describing TMB in general in our cohort in Figure 1. We have updated the corresponding result and method section to describe this more clearly.

Reviewer 2

Reviewer #2, expert in German clinical trials (Remarks to the Author):

--

Möhrmann and colleagues report on comprehensive and integrative analyses on genomic, transcriptomic and methylomic level from a subpopulation of patients with CUP syndrome enrolled into the MASTER program. The analysis population comprises 70 patients.

Patients with CUP syndrome have a poor prognosis; several attempts have been made to improve outcome e.g. by identifying site of origin. Besides non-comparative studies, one randomized trial has been reported aiming to better classify CUP patients regarding tumor site of origin and recommend respective treatment, however, as also discussed by the authors, this trial could not show that this approach actually leads to improved patient survival. Another randomized trial (SHIVA trial) compared treatment and outcome of patients with exhausted treatments options (not CUP patients) who received either standard of care or treatment as recommended by a molecular tumor board. The SHIVA trial was reported in 2016 and, unfortunately, could not show a substantial benefit for this approach either.

In brief, the manuscript adds to the existing body of evidence of translational diagnostics and treatment approaches for CUP syndrome or patients with exhausted treatment options. The overall number of patients is smaller compared to similar studies in the field and the patient population is likely not representative for CUP patients, because of the obvious bias towards younger patients. It provides some new aspects, and it reveals that such wealth and in-depth analyses can be contradictory and that there still remain many open questions who to translate such findings into patient care.

In the current version of the manuscript, there are several open issues that need to be addressed. This study is primarily a clinical study; thus, it should clearly report items as strongly recommended in reporting guidelines such as STROBE or REMARK. Especially regarding definition of analysis populations, calculation and reporting of clinical outcome parameters. I have the following specific comments that may help to improve clarity of the paper:

> We thank the reviewer for his valuable comments, which allowed us to improve our manuscript. We have improved our descriptions in order to clarify that we fulfill the reporting recommendations by guidelines like STROBE or REMARK.

1. Where there any general specific inclusion/exclusion criteria for the MASTER program? If yes, please report these criteria or provide them in the supplement. You refer to references 30 and 43, but this is not sufficient for the reader.

> We have added the following information under method section "NCT/DKTK MASTER": NCT/DKTK MASTER includes adults with advanced cancer across histologies who are younger than 51 years and patients with rare tumors, including rare subtypes of more common entities, regardless of age as cited in the manuscript. Patients must have exhausted curative treatment options and be in good general condition (Eastern Cooperative Oncology Group performance status of 0 or 1). Patients with cancers of unknown primary were included regardless of age due to its rarity.

2. Is the MASTER program registered with a clinical trial registry? If yes, please provide the respective registration identifier. If it was not registered, please provide the reason.

> We apologize for being unclear about this. NCT/DKTK MASTER is not a clinical trial and therefore not registered in a clinical trial registry. The MASTER program is a prospective multicenter observational study that provides a standardized diagnostic workflow which enables molecularly informed decisions for further therapy. Treatment recommendations are made in cooperation with treating oncologists following interdisciplinary discussion in a molecular tumor board. Depending on availability, further treatment could be given in clinical trials, off-label, or on-label. Within the reported cohort, none of the patients received targeted therapy within a registered clinical trial. This is partly because, unfortunately, the number of available clinical trials for CUP patients is very limited. We have updated the corresponding methods and result section accordingly.

3. Did you collect clinical patient data prospectively? You somehow describe this in section "Clinical and statistical analysis", but this is very superficial. How have clinical data been captured? Was there a centrally managed EDC system? Please comment.

> Clinical patient data was collected prospectively. Within the MASTER program we have implemented a centrally managed electronic data capture system (ONKOSTAR), which we have used to capture, organize and manage all clinical data in our cohort. At the time of inclusion, clinical data from doctor's letters, reports of imaging, histopathologic assessment and other diagnostic procedures were collected using a pre-specified data collection process. Afterwards, follow-up data on treatment and outcome of patients has been gathered for at least 24 months whenever possible. To address this more clearly, we have updated the corresponding method section accordingly.

4. What type of sample (core needle biopsy, resection of metastases) was required for participation in the MASTER program? Were analyses based on fresh biopsy at progression and / or at primary diagnosis? Please report on what types of samples you conducted the analyses.

> We used fresh frozen tumor specimens and matched normal control samples that were acquired at the time of inclusion in the MASTER program. Biopsies were obtained in the context of surgical procedures (25x resection) or by core needle biopsy (41x biopsy). Location of biopsy (as far as known) is listed in Supplementary Table 6. Supplementary Table 6 has been updated in order to address the question completely. Only in seven samples there were noteworthy exceptions as listed in Supplementary Table 6 (CUP-09: FFPE, CUP-14: bone marrow sample, CUP-17 and CUP-35: FFPE and already delivered as DNA extract, CUP-36: lumbar puncture sample; CUP-48: macrodissection for tumor cell enrichment; CUP-59: FFPE). The corresponding method part has been updated accordingly.

5. Please provide a statement on the type of design of your study. Given the information provided, I anticipate this is an observational prospective cohort study, but this needs clear statement in the paper.

> We have included the following part in the method section under “NCT/DKTK MASTER”:
“NCT/DKTK MASTER is a prospective, continuously recruiting, multicenter observational study.”

6. I cannot find any section commenting on written informed consent of patients and ethical review of the study. Especially regarding the germline analysis, patients usually have to provide additional written consent. Please add this to the manuscript.

> We have included the following part in the method section under “NCT/DKTK MASTER”:
“Patients provided written informed consent for banking of tumor and control tissue, molecular analysis including germline analysis, and the collection of clinical data under a protocol (S-206/2011) approved by the Ethics Committee of the Medical Faculty of Heidelberg University. The study was conducted in accordance with the Declaration of Helsinki.”

7. You obviously used the program R for the bioinformatical part, but I cannot find a description of the statistical program you used to calculate the clinical outcome parameters such as PFS and response. Please comment on this and provide description in the methods section.

> Since our clinical analysis relied on basic mathematical calculations, we used Microsoft Excel and the R script ggplot2 to facilitate this process. We prepared tables similar to our supplementary tables to organize clinical data and calculate relevant parameters such as PFS. We have added the following phrase under the method section “Additional data processing and analysis”:
“PFS, PFSr and mPFSr were calculated using Microsoft Excel. Survival analysis using Kaplan-Meier estimator and log-rank tests was performed using ggplot2 (version 3.3.3). P values < 0.05 were considered statistically significant.” Minor mistakes were corrected in the manuscript.

8. It is not clear to me whether reported patients were included from one or several centers. You mention the DKTK consortium, but this is not really clear for readers not being familiar with current structures in Germany. Was this a single or multicenter study? If multicenter, from how many centers were patients included?

> All DKTK sites are actively recruiting patients within the MASTER program. We have added the following information under the method section “Clinical and statistical analysis”:
“Clinical data for cohort selection, description and analysis was obtained from the National Center for Tumor Diseases (NCT) Heidelberg and Dresden as well as from six other comprehensive cancer centers (CCCs) of the German Cancer Consortium (DKTK). The DKTK network includes ten CCCs at eight sites (Berlin, Dresden, Essen/Düsseldorf, Frankfurt/Mainz, Freiburg, Heidelberg, Munich, Tübingen).”

9. Where have the analyses been conducted? Was it one laboratory or several sites? If yes, how was standardization of assays etc. guaranteed?

> All molecular analyses were done centrally in Heidelberg using the well-established and standardized clinical genomics workflow of the MASTER program. All samples underwent processing by the NCT Molecular Diagnostics Program Sample Processing Laboratory (SPL) and sequencing

analysis by the DKFZ Genomics and Proteomics Core Facility (GPCF). Detailed information concerning test kits and quality control is listed in the method section as well as in the supplements. We have updated the corresponding method section accordingly.

10. Did you pre specify tests and panels to be done? E.g. a minimal core set of WES followed by other techniques / analyses if sufficient tissue was available?

> From May 2013 to October 2016 whole exome sequencing (WES) was the standard sequencing method within the MASTER program. Since November 2016 we performed whole genome sequencing (WGS) instead of WES in a substantial proportion of patients. If RNA quality was sufficient, RNAseq was performed as described in the manuscript. If tumor cell content was >30%, methylation analysis was performed as described in the manuscript but methylome information was not available for the molecular tumor board since it was generated later than DNA-/RNAseq data. We have described this more clearly in the manuscript and updated Supplementary Table S15.

11. Regarding the entity prediction, were results from these analyses available to the MTB when discussing the cases and making recommendations?

> We apologize for not being precise enough when describing this part in the manuscript. Entity predictions were calculated retrospectively and therefore not available for treatment recommendations by our molecular tumor board. In the result section “Entity prediction using methylome and transcriptome data” we have added the sentence:

“We retrospectively performed entity predictions using methylome- and transcriptome-based similarity analysis.”

In the discussion we have added the sentence:

“In our cohort, MTB recommendations were not influenced by our methylome- and transcriptome-based entity prediction since it was not available at the time of the MTB.”

12. The reported 70 patients were relatively young and are likely not representative for patients usually diagnosed and treated with CUP syndrome. How do you explain this? May this be because of your inclusion/exclusion criteria?

> We agree with the reviewer, as also addressed in our discussion; the reported patients were younger than we would expect them to be in a representative cohort based in Germany (Kraywinkel und Zeissig, 2017; abstract available in English, <https://doi.org/10.1007/s00761-017-0301-z>). NCT/DKTK MASTER includes patients between age 18 and 50 and/or patients with rare cancer types such as CUP regardless of age. This might have influenced oncologists to particularly refer younger CUP patients to our program. Additionally, we only included patients who had exhausted curative treatment options and were in good general condition (Eastern Cooperative Oncology Group performance status of 0 or 1). Younger patients are more likely to fulfil these criteria. We have updated the corresponding discussion part of the manuscript.

13. You report to have identified and included 70 patients with CUP syndrome, however, only 61 patients fulfilled the ESMO criteria. I assume this is caused by discrepancy between inclusion criteria of the MASTER program and the ESMO criteria. Why is this? Please comment on this.

> As stated in the result part of the manuscript, the documentation of the initial imaging procedures (such as CT scans of thorax, abdomen and pelvis) was not available in 9 patients. According to ESMO guidelines these are required for the diagnosis of CUP syndrome. Nevertheless, these patients were enrolled in the MASTER program since we relied on the diagnosis of the referring treating oncologists. We have updated the corresponding method section to address this more clearly.

14. Please provide percentage in addition to the absolute numbers in table 1.

> We have provided an updated table with percentages.

15. One of most central major revisions required is the definition of analysis populations and outcome reporting. Current presentation of clinical outcome is certainly biased.

> We refer to our answer to comment 16.

16. Specifically: First, all 70 patients should be in the denominator when reporting outcome, thus, for all 70 patients PFS and OS should be calculated and reported. Also, for all these 70 patients you should calculate the PFS ratio irrespective of the MTB recommendation or whether recommended treatment has been given. Second, if you want to explore potential associations of benefit between personalized treatment and outcome, you need at least to contrast outcome (PFS, response, PFS ratio, OS) of patients who received the recommended treatment (N=20), but also the outcome of those patients who had a recommendation but did not receive the recommended treatment (N=36). A stratified table, similar to content of table 1, would be helpful allowing to compare patient and disease characteristics. In addition, information on the evidence level of the MTB recommendation would also enrich the content. However, such exploratory analysis would still not allow to conclude that personalized treatment based on an MTB leads to improved patient outcome, because of several unmeasured confounding factors. For such statement you need a randomized trial. This limitation also needs to be addressed in the discussion.

> We thank the reviewer for these important remarks. It is true that our study is not a randomized trial and we listed this topic in the revised manuscript more clearly among the limitations of our study that we have already addressed in the discussion.

We followed the approach of PFS2/1 ratios in patients receiving genomics-based therapies that was initially described by Von Hoff et al. J Clin Oncol 2010 (<https://doi.org/10.1200/jco.2009.26.5983>). In this analysis each patient is his/her own control comparing the molecularly informed treatment (PFS2) with the last systemic treatment before molecular analysis (PFS1). This approach has also been used in various published molecular profiling studies such as the MOSCATO or WINTHER trial which are cited in the manuscript. Nevertheless, we have added the clinical outcome of patients that did not receive a recommended targeted treatment as proposed by the reviewer. We have added the corresponding PFS as well as OS and response data of this cohort as requested. We have divided the clinical results into the chapters “Genomics-based treatment recommendations” and “Genomics-based systemic treatment” in order to keep it well-arranged for the reader.

Furthermore, we provide a stratified table as requested (Supplementary Table S13). We included NCT/DKTK evidence levels and further outcome parameters in the corresponding result sections: “All drug recommendations were sorted into groups with the evidence level they were based on (Level 1 A/B/C, 11/142, 8%; Level 2 A/B/C, 89/142, 63%; Level 3, 31/142, 22%; Level 4: 11/142, 8%).” “The distribution of evidence levels of clinically applied recommendations showed a similar distribution as the one of all drug recommendations (Level 1, 2/20, 10%; Level 2, 15/20, 75%; Level 3, 1/20, 5%; Level 4, 2/20, 10%).”

“Of 20 patients who received the recommended targeted therapies, 12 (60%) had stable disease ≥ 6 months or achieved objective remissions (PR/CR).”

“For patients that did not receive a recommended treatment, we calculated the ratio of the first treatment applied after the MTB (PFS_b, n = 12) and the last prior systemic treatment (PFS_a, n = 11) which resulted in a mean PFS_r of 0.67 (median PFS_r = 0.71, range 0.1 to 1.0, n = 11; Supplementary Table S14). Median overall survival of the 36 patients without application of recommended treatments was significantly shorter than of the 20 patients that received a recommended therapy (18.3 months vs. 34.8 months, p < 0.05; Supplementary Figure S4). Same was true for median PFS2 and PFS_b (7.8 months vs. 3.8 months, p < 0.001; Supplementary Figure S4). Two patients with PFS_b had stable disease ≥ 6 months or achieved objective remissions (PR/CR; Supplementary Figure S4). Since our study was not randomized, these results are not controlled for possible confounding factors. Further data are provided in Supplementary Table S13 and S14.”

In the discussion we have added the sentence:

“Third, our study was not a randomized clinical trial but a prospective observational study.”

17. For all Kaplan-Meier plots, please provide 95% confidence intervals and at-risk table below the graph as this is standard in reporting time-to-event data.

> We have updated all Kaplan-Meier plots providing 95% confidence intervals and number at risk tables below the graph (Supplementary Figure S4).

18. Please report the absolute number of PFS defining events together with PFS. Please also report OS and the absolute number of deaths. Finally, also report the follow-up time of all patients, e.g. using the inverse Kaplan-Meier method.

> Of 20 patients that received molecularly-guided therapy, 19 showed progressive disease and one death as PFS defining event. The corresponding result and method section have been updated. The absolute number of deaths during the observation period in the whole cohort of 70 patients was 38, we have introduced this information in the result section "Patients". Median OS of those 38 patients is 12.0 months as reported in supplementary table 1. Follow-up time of all patients using the inverse Kaplan-Meier Method has been added in Supplementary Figure S4 as suggested by the reviewer. Median follow-up time (25.2 months) and median OS (22.1 months) have been added to the result section "Patients".

19. How was treatment in accordance with the MTB recommendation defined? Please specify this.

> Only systemic treatments that matched with at least one of our molecular tumor board recommendations for the given patient and that were initiated after inclusion into the MASTER program were considered as treatment in accordance with MTB recommendations. We have updated the method section with the sentence:
"Progression-free survival (PFS) of the first applied treatment recommended by the MTB (PFS2) was compared to the PFS of the last prior systemic treatment (PFS1) in each individual patient."

20. Where there any pre-specified assessments or where patient outcomes determined based on routinely collected data? Please specify.

> Patient outcomes were determined by centrally and routinely collected clinical data based on doctors' letters and diagnostic reports including histopathological assessments or imaging procedures at baseline and during follow-up. All clinical data were reviewed and assessed by a team of medical curators on the basis of pre-specified rules for a harmonized interpretation. We have updated the corresponding method section in order to describe this more clearly. We are aware of the limitations of observational studies. Nevertheless, our observational data provide valuable and clinically important real-world data on the impact of genomics-guided therapies in CUP patients and may pave the way for corresponding clinical trials to confirm the efficacy of molecularly stratified treatment approaches in this entity.

21. How was PFS calculated? Please provide the starting date for calculating the PFS as well as definition of PFS defining events.

> In each individual case, the starting date of a given PFS was the first day of drug application. Due to data protection guidelines we are not allowed to provide individual starting dates in the manuscript. End of PFS was either death or progressive disease. We have updated that definition in the manuscript.

22. How did you collect information PFS of the most previous therapy before inclusion into the MASTER program?

> PFS1 was assessed the same way as described in our answer to question 21. PFS started with the first day of therapy application and ended with documented disease progression. As mentioned above, all data were reviewed by a team of curators.

23. Have you collected the reasons why recommended treatments were not applied? Was this a reimbursement issue? This is a very central issue in all initiatives of personalized oncology treatment. Please comment.

> Within our MASTER trial, reasons for non-implementation of MTB recommendations included, e.g. worsening of patient's general condition, death before treatment could be given, and lack of access

to or reimbursement of the recommended drug(s) as described in <https://doi.org/10.1158/2159-8290.cd-21-0126>. We have updated the corresponding result part with this information. More detailed information has been introduced in Supplementary Table 14.

24. Please report all outcomes in months, not in days.

> We converted all outcome data to months as requested by the reviewer (please see manuscript and Supplementary Table S12 in particular).

Reviewers' Comments:

Reviewer #1:

Remarks to the Author:

This is a second review of the manuscript prepared by Möhrmann and colleagues. I think the manuscript has been improved and I praise the authors for the extra effort. I can accept most of the responses made to my initial review questions. However, I do feel there are some parts that could still be improved further to better describe the data and analysis done.

Molecular characteristics

1. May I suggest the authors consider adding some further detail in Figure 1 where new information has come to light as this is relevant to the molecular landscape of the CUP cohort as well as diagnosis and treatment decisions made
 - Consider replacing "HRD" with "LOH+HRD + LST" in the Figure to differentiate from the other methods used to infer HRD. Also consider adding annotation for CHORD and HRDetect where the score was above the threshold prescribed for calling HRD in the original papers describing these methods.
 - It would also be useful to add any data relating to mutated HRD-related genes (e.g BRCA1, CHEK2)
 - Add dominant mutational signatures AC4 and AC7 to the Figure.
 - Add annotation where HPV sequence was detected as these have important diagnostic relevance.
 - The authors could consider adding a histology annotation key in the figure. For instance, one might like to know if HPV was detected in SCCs.
 - Include data for cancer genes associated with homozygous deletion (e.g. CDKN2A n=7) as this is an important feature like fusion events, SNVs and InDels. HD of CDKN2A is mentioned even in the abstract so it seems appropriate to include in the Figure?

Germline analysis

2. State whether somatic LOH was concordant with pathogenic germline mutations. Germline mutations with concordant LOH could also be added to Fig 1 as above.
3. Line 214. There is an incorrect callout of Supplementary Table 4.1 (should be Supp Table 5.1)
4. I note that the heterozygous germline FH variant c.1431_1433dupAAA has recently been shown not to be associated with increased risk of developing fumarate deficient HLRCC
<https://pubmed.ncbi.nlm.nih.gov/31444830/>

Gene-expression and DNA Methylation Classification

5. The authors have now tested a modest number of tumors from the MASTER set using the TCGA trained classifiers. Previous studies have shown that gene-expression and DNA methylation tissue of origin classifiers can achieve test accuracies for metastatic tumors in the order of 80-90%. Could it be that the poorer performance (~49-56%) observed using the three TCGA trained RNA classifiers (incl. CUP-AI-Dx, CancerScope and the authors own method) relates to a systematic bias existing between MASTER and TCGA datasets? For instance, the TCGA set is made of primary cancers, whereas the gene-expression classifiers are now being tested on metastatic cancers? This caveat should be acknowledged in interpreting the results and making any conclusions about the relative performance of these methods on the MASTER CUP cohort.

6. Line 260 Written "This approach showed the best accuracy when tested on our validation cohort (54/72, 75%) and enabled us to have a comparison to rare tumor entities which were enrolled in MASTER but are not part of TCGA". Please provide more information on these additional rare cancer types represented in the MASTER series and not in TCGA. The authors may also need to consider whether the reference transcriptome data for the MASTER cohort needs to be made available for reproduction of the classification results. Would this not be a requirement of the journal? I tried

accessing the EGA accession cited in the manuscript but I could not see what data has been uploaded.

7. Regarding the method used for RNA classification. How did the authors arrive at the arbitrary threshold of FPKM > 5 for the RNA classifier? Was there any tuning using a leave one out cross validation of the training dataset? As per my in comments from initial review a LOOCV and tuning of a classifier is standard practice as it functions to tune the classifier but also show the relative performance can be maintained between training and independent test sets.

8. Line 249-250 Written "When comparing TCGA cohorts with our CUP cohort, classification based on methylome comparison led to an entity prediction in 55/70 cases (79%)." I find this sentence a little misleading as it suggests that the classifier could only make a prediction in 55 cases but really what is meant is that prediction was only attempted in 55 cases. I feel this section could be written in a more concise manner.

9. Line 264 Written "It highlights the need for an integrated classifier taking into account both methylation and transcriptomic data (Figure 2)". I think the description now provided in the legend for Figure 2 including a justification for the tSNE plots and the interpretation of the data should be in the main body of the text. The authors should also perhaps reconsider the call out of tSNE plots in panels 2A and 2B. One suggestion is to call out (2A, 2B) at the very beginning of the Results section titled "Entity prediction using methylome and transcriptome data". Something general could be said about the heterogeneity of the CUPs based on DNA methylation (GEP) profile, that CUPs cluster among a range of solid cancers or carcinomas but to the exclusion of some cancer types. This could then be a segue to describing the multi-class classification methods. I think for completeness a tSNE or UMAP of transcriptome data should also be shown for comparison to methylation. I will also add that other than the reader getting the impression from tSNEs that CUPs are heterogeneous and cluster among a range of cancer types the plots are rather impossible to interpret given the large number of cancer types and spectrum of colors in the cancer type key. For instance, the comment that CUPs cluster with PAAD and CHOL is hard to see in the plot. Perhaps the authors could consider improving this figure by embedding class labels proximal to clusters within the tSNE itself or at least highlighting some of the major tumor clusters within the plot so the reader can easily interpret the data.

10. Panel 2C. Total cases n=55 is shown in the non-overlapping portion of the Venn circles for all classifiers. My understanding is that the convention in a Venn diagram would be to show the number of non-concordant cases in the outer non-overlapping part of the circle. Furthermore, the number of cases in the direct comparison between gene-expression and methylation is 48 not 55 as written. Therefore the number in the non-overlapping part of the circle is 48 minus the number of cases overlapping with either of the other two classification methods. A separate panel should be made for direct comparison of the gene-expression classifiers as the number of cases tested is different.

Reviewer #2:

Remarks to the Author:

The authors have sufficiently addressed my previous comments. I still have some final suggestions:

Regarding "Statement of Significance", the conclusions are too strong regarding the clinical benefit. I strongly suggest tuning down the wording, e.g. "...targeted therapies showed some clinical activity, but definitive clinical benefit of this approach requires further evaluation in dedicated clinical trials."

In the chapter "Genomic based treatment recommendations", please also mention here that only sequencing data were available to MTB when discussing patients and making treatment recommendations.

Please also provide the turn-around times from testing to first discussion of the patients at the MTB.

This time (median, range) should also be shown in table S13, stratified and in total.

Table S13: The variable "MTB recommendations" does not sum up to the denominator of 56 patients; there should not be two units of investigations in one table. Also, please provide a total column for the column.

Reviewer's #1 comments

This is a second review of the manuscript prepared by Möhrmann and colleagues. I think the manuscript has been improved and I praise the authors for the extra effort. I can accept most of the responses made to my initial review questions. However, I do feel there are some parts that could still be improved further to better describe the data and analysis done.

> We thank the reviewer for his positive response to our revised manuscript. In the following, we would like to address each comment point by point.

Molecular characteristics

1. May I suggest the authors consider adding some further detail in Figure 1 where new information has come to light as this is relevant to the molecular landscape of the CUP cohort as well as diagnosis and treatment decisions made

- Consider replacing “HRD” with “LOH–HRD + LST” in the Figure to differentiate from the other methods used to infer HRD. Also consider adding annotation for CHORD and HRDetect where the score was above the threshold prescribed for calling HRD in the original papers describing these methods.

> We have added this information to Figure 1 as requested.

- It would also be useful to add any data relating to mutated HRD-related genes (e.g BRCA1, CHEK2)

> We introduced an additional annotation bar in the Figure 1, which shows the presence of SNVs, indels or fusions in HRD-related genes. To do so, we queried 12 HRD-related genes (Supplementary Table S17.1) in which mutations were previously observed [Mateo et al. 2015, NEJM, DOI: 10.1056/NEJMoa1506859]. Details about mutations underlying the annotation were included in an additional supplementary table.

- Add dominant mutational signatures AC4 and AC7 to the Figure.

> We have added this information to Figure 1 as requested.

- Add annotation where HPV sequence was detected as these have important diagnostic relevance.

> We have added this information to Figure 1 as requested.

- The authors could consider adding a histology annotation key in the figure. For instance, one might like to know if HPV was detected in SCCs.

> We have added this information to Figure 1 as requested.

- Include data for cancer genes associated with homozygous deletion (e.g. CDKN2A n=7) as this is an important feature like fusion events, SNVs and InDels. HD of CDKN2A is mentioned even in the abstract so it seems appropriate to include in the Figure?

> In our study, we identified focal somatic copy number alterations in CUP patients. We agree with the reviewer that the homozygosity of deletions is interesting and could be presented in our already comprehensive Figure 1. In our analysis, we define focal event as a distinct copy-number segment spanning less than 3 Mb. Such a threshold allows for recognition of almost all homozygous deletions [Cheng et al. 2017, <https://www.nature.com/articles/s41467-017-01355-0>]. However, their reliable detection can be challenging. The two pipelines that we use for CNV analysis, due to the nature of the data (WES and WGS), perform differently in that respect causing under-calling (CNVkit) or over-calling (ACEseq) of homozygous deletion events. To overcome the over-calling problem, we focus on genes that are relevant to cancer, namely those present in Cancer Gene Census. We added these results to Figure 1. In short, the most frequent (n>1) focal homozygous deletions

observed in Cancer Gene Census genes are: *CDKN2A* (n=6), *FHIT* (n=4), *ROBO2* (n=3), *CCR7* (n=2), *FOXP1* (n=2), *LRP1B* (n=2). In the previous version of manuscript, we wrote that we observe focal deletion of *CDKN2A* in seven patients, with five being homozygous and two - hemizygous. After deeper look into the results we reclassified one hemizygous patient to the homozygous group, because although most part of *CDKN2A* gene in this patient is indeed a hemizygous deletion, there is a small 230 bp long segment that is classified as homozygous deletion. Therefore, we modified the text accordingly. Moreover, we have improved the clarity of our description: adding a word “homozygous” in the abstract and introducing the definition of focal copy number alteration in the methods.

Germline analysis

2. State whether somatic LOH was concordant with pathogenic germline mutations. Germline mutations with concordant LOH could also be added to Fig 1 as above.

> The only patient with a (likely) pathogenic germline variant and concordant LOH is CUP-64 who has a *CDKN2A* germline variant. Since it is only one patient, we did not add this information to Figure 1 but introduced the information in the corresponding result part in the manuscript. We added LOH annotation in the respective supplementary table.

3. Line 214. There is an incorrect callout of Supplementary Table 4.1 (should be Supp Table 5.1)

> We have verified that all tables and figures are referred to correctly in the manuscript. While doing that we could not confirm an incorrect callout in line 214.

4. I note that the heterozygous germline FH variant c.1431_1433dupAAA has recently been shown not to be not associated with increased risk of developing fumarate deficient HLRCC <https://pubmed.ncbi.nlm.nih.gov/31444830/>

> We thank the reviewer for bringing this to our attention. We agree that this variant should rather be assessed as variant of unknown significance in regards to cancer predisposition. Therefore, we have updated the assessment in the respective supplementary table.

Furthermore, we have changed the description of our 101 gene list used for filtering of rare germline variants to “cancer predisposition genes” in the text as well as in the supplement in order to be more concise.

Gene-expression and DNA Methylation Classification

5. The authors have now tested a modest number of tumors from the MASTER set using the TCGA trained classifiers. Previous studies have shown that gene-expression and DNA methylation tissue of origin classifiers can achieve test accuracies for metastatic tumors in the order of 80-90%. Could it be that the poorer performance (~49-56%) observed using the three TCGA trained RNA classifiers (incl. CUP-AI-Dx, CancerScope and the authors own method) relates to a systematic bias existing between MASTER and TCGA datasets? For instance, the TCGA set is made of primary cancers, whereas the gene-expression classifiers are now being tested on metastatic cancers? This caveat should be acknowledged in interpreting the results and making any conclusions about the relative performance of these methods on the MASTER CUP cohort.

> The reasons for the relatively poor performance are multifactorial. The overrepresentation of metastatic disease in the validation cohort and the lack thereof in the TCGA reference cohort are probably an important factor, as suggested by the reviewer. Another likely reason is that the MASTER cohort by design comprises rare cancer types, thus impairing the match-making between MASTER and TCGA patients. Moreover, we note that a number of misclassifications concern cancer types with transcriptionally similar cells of origin. For example, frequent misclassifications were made between cholangiocellular (CHOL),

pancreatic (PAAD), and hepatic cancer (LIHC), presumably owing to a common site of metastasis and possibly also transcriptionally similar cells of origin. Likewise, uveal melanoma (UVM) and skin cutaneous melanoma (SKCM) were also swapped in several cases. While these lesions may be straightforward to distinguish using external information (e.g., anatomical site of lesion), the classification exclusively based on transcriptomic data is challenging. Lastly, technical differences in the library preparation protocols and sequencing protocols of MASTER and TCGA certainly contribute to the misclassification rate as well, since expression quantification using RNA-Seq is known to be sensitive to such protocol differences. We have incorporated these thoughts into the discussion.

6. Line 260 Written “This approach showed the best accuracy when tested on our validation cohort (54/72, 75%) and enabled us to have a comparison to rare tumor entities which were enrolled in MASTER but are not part of TCGA”. Please provide more information on these additional rare cancer types represented in the MASTER series and not in TCGA. The authors may also need to consider whether the reference transcriptome data for the MASTER cohort needs to be made available for reproduction of the classification results. Would this not be a requirement of the journal? I tried accessing the EGA accession cited in the manuscript but I could not see what data has been uploaded.

> MASTER particularly focuses on young patients and patients with rare cancers. Therefore, MASTER includes entities that are not part of TCGA. Examples are DSRCTs, adenoid cystic carcinomas and several types of sarcomas. We have added a complete list of diagnoses that are part of the reference cohort in an additional supplemental table. Additionally, we have uploaded the expression values of the reference cohort and the validation cohort as a matrix of FPKM values to EGA. Moreover, the methylation beta values of the validation cohort have been added as well.

7. Regarding the method used for RNA classification. How did the authors arrive at the arbitrary threshold of $FPKM < 5$ for the RNA classifier? Was there any tuning using a leave one out cross validation of the training dataset? As per my in comments from initial review a LOOCV and tuning of a classifier is standard practice as it functions to tune the classifier but also show the relative performance can be maintained between training and independent test sets.

> The thresholds were defined based on biological rationale. The gist of the algorithm is the selection of marker genes that distinguish samples/tissues from each other. To this end, genes are selected which are expressed in one tissue but not in the other. A threshold of < 0.5 FPKM was assumed to select genes that are not expressed, and a ten-fold higher threshold (> 5 FPKM) was assumed to be sufficiently larger to select stably expressed genes.

We followed the reviewer’s advice and performed 10-fold cross-validation on a subset of the reference cohort, which served as a training set. The following heat map shows the prediction accuracies across the explored parameter space. We evaluated all integer combinations between 0 and 20 FPKM for the two thresholds for non-expressed and highly expressed genes. The procedure identified the combination of 3 FPKM for non-expressed genes and 13 FPKM for highly expressed genes to be the optimum.

Next, we applied the optimized parameters to the validation cohort, which confirmed that the new parameters indeed improve the predictions (three more correct predictions than with the non-optimized parameters).

Therefore, we have updated the CUP predictions in the manuscript based on the new thresholds.

8. Line 249-250 Written “When comparing TCGA cohorts with our CUP cohort, classification based on methylome comparison led to an entity prediction in 55/70 cases (79%).” I find this sentence a little misleading as it suggests that the classifier could only make a prediction in 55 cases but really what is meant is that prediction was only attempted in 55 cases. I feel this section could be written in a more concise manner.

> We have rewritten this sentence, the wording has been changed to “classification ... was possible.”

9. Line 264 Written “It highlights the need for an integrated classifier taking into account both methylation and transcriptomic data (Figure 2)”. I think the description now provided in the legend for Figure 2 including a justification for the tSNE plots and the interpretation of the data should be in the main body of the text. The authors should also perhaps reconsider the call out of tSNE plots in panels 2A and 2B. One suggestion is to call out (2A, 2B) at the very beginning of the Results section titled “Entity prediction using methylome and transcriptome data”. Something general could be said about the heterogeneity of the CUPs based on DNA methylation (GEP) profile, that CUPs cluster among a range of solid cancers or carcinomas but to the exclusion of some cancer types. This could then be a segue to describing the multi-class classification methods. I think for completeness a tSNE or UMAP of transcriptome data should also be shown for comparison to methylation. I will also add that other than the reader getting the impression from tSNEs

that CUPs are heterogeneous and cluster among a range of cancer types the plots are rather impossible to interpret given the large number of cancer types and spectrum of colors in the cancer type key. For instance, the comment that CUPs cluster with PAAD and CHOL is hard to see in the plot. Perhaps the authors could consider improving this figure by embedding class labels proximal to clusters within the tSNE itself or at least highlighting some of the major tumor clusters within the plot so the reader can easily interpret the data.

> We have added tSNE plots based on the transcriptome data as requested (Figure 2C+D). Additionally, we have also rearranged the manuscript and put part of the figure legend into the result section as requested by the reviewer.

However, the suggested embedding of class labels creates an even more complex plot and most of the labels in the middle of the methylome plot highly overlap (for illustration see eclipse representation of regions by entity in the tSNE plot attached). For this reason, it will not be possible to highlight just some clusters to better interpret the CUP samples within the methylome plot. We have tried this before but there is no way that by means of a tSNE plot one can easily and correctly assign the individual CUP points to entity clusters just by eyeballing. The same problem occurs with the transcriptome tSNE plot although the overlap is slightly less severe. We would further argue that this is not the purpose of the tSNE plot. The reader who is interested in the detailed stats can refer to the supplemental tables that contain helpful information. Figure 2A-D is intended to only illustrate the heterogeneity of CUP samples and which entity clusters they roughly align to in these two dimensions.

10. Panel 2C. Total cases $n=55$ is shown in the non-overlapping portion of the Venn circles for all classifiers. My understanding is that the convention in a Venn diagram would be to show the number of non-concordant cases in the outer non-overlapping part of the circle. Furthermore, the number of cases in the direct comparison between gene-expression and methylation is 48 not 55 as written. Therefore the number in the non-overlapping part of the circle is 48 minus the number of cases overlapping with either of the other two classification

methods. A separate panel should be made for direct comparison of the gene-expression classifiers as the number of cases tested is different.

> We have included a new version of figure 2C (now called Figure 2E). As suggested, we corrected the number of total comparable cases to 48 and replaced the case-counts inside non-overlapping circles with respective numbers of non-concordant cases. Furthermore, we have provided a separate panel for direct comparison of the gene-expression classifiers as proposed by the reviewer (Figure 2F).

Reviewer #2 (Remarks to the Author):

The authors have sufficiently addressed my previous comments. I still have some final suggestions:

Regarding “Statement of Significance”, the conclusions are too strong regarding the clinical benefit. I strongly suggest tuning down the wording, e.g. “...targeted therapies showed some clinical activity, but definitive clinical benefit of this approach requires further evaluation in dedicated clinical trials.”

> We have updated the wording according to the reviewer’s wishes.

“Statement of Significance” - old version:

Treatment options for CUP patients are limited and often ineffective resulting in a dismal prognosis. We demonstrate that a comprehensive precision oncology approach provides clinically relevant information and additional, molecularly stratified treatment options in many cases. The majority of these targeted therapies was highly beneficial even in heavily pretreated patients.

“Statement of Significance” - new version:

Treatment options for CUP patients are limited and often ineffective resulting in a dismal prognosis. We demonstrate that a comprehensive precision oncology approach provides clinically relevant information and additional, molecularly stratified treatment options. Targeted therapies showed clinical activity even in pretreated patients warranting further evaluation in dedicated clinical trials.

In the chapter “Genomic based treatment recommendations”, please also mention here that only sequencing data were available to MTB when discussing patients and making treatment recommendations.

> We have introduced this information in the first sentence of the respective result chapter as recommended by the reviewer.

Please also provide the turn-around times from testing to first discussion of the patients at the MTB. This time (median, range) should also be shown in table S13, stratified and in total.

> We have added the turn-around times from testing to MTB in Supplementary Table 1 and mention the median value (and range) in the result section as well as in the corresponding supplementary table as requested.

Table S13: The variable “MTB recommendations” does not sum up to the denominator of 56 patients; there should not be two units of investigations in one table. Also, please provide a total column for the column.

> We apologize for this mistake. We have updated the supplementary table as requested.

Reviewers' Comments:

Reviewer #1:

Remarks to the Author:

The authors have now adequately addressed all of my comments

Reviewer #2:

Remarks to the Author:

The authors have sufficiently answered/edited all my requests, no more comments.

NCOMMS-21-08986A (Major Revision)

> We thank the reviewers for their valuable comments. We have addressed comments as requested which has profoundly improved the quality of our manuscript. Below you will find our answers to each comment.

Reviewer 1

Reviewer #1, expert in genomics for precision oncology (Remarks to the Author):

Möhrmann and colleagues have described a multi-omics analysis of cancers of unknown primary. The study describes a) frequencies of somatic and pathogenic germline gene mutations across the cohort b) comparison of RNA and DNA methylation based tissue of origin classifiers c) details of some potential diagnostic mutational features and d) description of clinical outcome in a subset of patients given genomics directed treatments. The cohort size is modest (n=70) compared to other previously published series using panel DNA sequencing (e.g. JAMA Oncol 2015 Vol. 1 Issue 1 Pages 40-9 (n=200) , Ann Oncol 2017 Vol. 28 Issue 12 Pages 3015-3021 (n=150)). The reported frequency of specific gene mutations is similar to that reported in prior studies and few conclusions can really be made from the data other than CUP is a molecularly heterogeneous disease, which is already well known. It may also be questionable if the series is truly representative of CUP given the relatively young average age of the patients and inclusion of non-carcinoma types including melanoma and small round cell tumors. Contrasting results from two orthogonal molecular classifiers is one very interesting aspect of the study but was quite poorly executed. Similarly, the interpretation and reporting of mutational signatures from WGS and WES data could have been done better and in this regard the study fails to harness one of the major benefits of taking a genome-wide and multi-omics approach. In summary, I felt the study could have been done better. As it stands it does not really provide any major insights into our understanding of CUP biology or highlight the application multi-omics analysis of CUPs using current "best practice" methods. As a clinical report I think it is still an important study highlighting the importance of molecular profiling for clinical decision-making in CUP.

1. Mutational signature analysis. The description and interpretation of COSMIC mutational signatures needs refinement. For example, a UV signature is described in 12 cases which may imply a potential skin origin- an unusually high fraction of such cases among CUP tumors. A UV signature is usually detected in the context of high mutational burden but this does not appear to be the case in the majority of CUPs given only three tumors in the whole series have TMB >10muts/Mb. Was the UV signature dominant in all cases where it was detected? Similarly, smoking sig4 is described in 9 case in the manuscript but is only noted as a useful feature in 3 cases in Supp Table 5. If the authors are confident there is a dominant mutational signature such as UV (sig 7) or smoking (sig 4) then these features have important diagnostic utility and this information should be presented in an interpretable way.

> We have extended the presentation of results on mutational signature analysis for signatures with diagnostic potential, namely AC3, AC4, AC7 and AC11. First, we included the number of patients per dominant signature in the main text. Secondly, we added figures S3 b-e from which the reader can not only see the dominance level of each relevant signature in samples in which it was detected, but also both non-normalized and normalized exposure values. Where applicable, information on

dominant signatures was added to the results for tissue origin prediction included in Supplementary Table 6.

Nonetheless, it needs to be considered that mutational signatures corresponding to ultraviolet light exposure and tobacco smoking have been previously described in cancers of origin outside skin and lung, respectively (AC7:

https://cancer.sanger.ac.uk/signatures/media/images/v3.2_SBS7a_TISSUE.original.jpg,

https://cancer.sanger.ac.uk/signatures/media/images/v3.2_SBS7b_TISSUE.original.jpg,

https://cancer.sanger.ac.uk/signatures/media/images/v3.2_SBS7c_TISSUE.original.jpg,

https://cancer.sanger.ac.uk/signatures/media/images/v3.2_SBS7d_TISSUE.original.jpg,

AC4: https://cancer.sanger.ac.uk/signatures/media/images/v3.2_SBS4_TISSUE.original.jpg).

This data has been published by Alexandrov et al. in Nature, 2020 (<https://doi.org/10.1038/s41586-020-1943-3>).

Moreover, based on that data, both signatures have been frequently identified in samples with TMB < 10mut/Mb. Therefore, we are careful with the interpretation of mutational signature analysis results for the diagnostic purposes in CUPs. However, we greatly appreciate the reviewers' suggestion to focus on signature dominance level. For example, in samples with dominant signature AC7, tissue of origin was unanimously predicted by all tools tested by us to be skin. Signature dominance is a valid and important clue about the tissue of origin, but in order to be a strong one, we believe, it still should be accompanied by more evidence. We have also updated the corresponding discussion part.

2. HRD prediction. With regards to HRD the authors reported LOH and LST patterns with combined combined scores as well as sig3. Were both sig 3 and LOH/LST features used together to interpret HRD for directing PARPi? In Supp Table 6.1 it appears that those cases with reported sig 3 had low or medium LOH/LST scores (e.g. CUPs 54 and 66) yet these cases also had BRCA2 mutations. More reliable methods for predicting HRD in pan-cancer data have been developed using WGS and these methods are superior to genomic scar-based indexes or sig3 alone. These newer methods include HRDetect (Nat Med 2017 Vol. 23 Issue 4 Pages 517-525) and CHORD (Nat Commun 2020 11, 5584. CHORD is freely available (<https://github.com/UMCUGenetics/CHORD>))

> We thank the reviewer for pointing out these additional methods that have been published recently. We have added both of them, HRDetect and CHORD. They have enriched our analysis and allowed us to correlate their results with our method that is being used for the molecular tumor board in MASTER. Both HRDetect and CHORD predict HRD in WGS samples, whereas more than half of our cohort data comes from WES. Results of both methods are in line with our method. When discussing patients in the molecular tumor board we depend on methods that can be applied broadly to as many patients as possible. Therefore, we have reported our method as it was used in the molecular tumor board. HRD was interpreted based on a summary of signature AC3, LOH/LST scores, somatic and germline mutations and used as an argument for recommending PARPi.

3. Tissue of origin classification. A major deficiency here was that the performance of the classifiers was not assessed using an independent set of metastatic tumors of known origin. This is an essential step in developing and validating a ToO classifier otherwise overfitting of the data and/or technical biases between training and test sets cannot be ruled out. Classifying a tumor based on the nearest paired tumor sample in the reference by Spearman correlation really lacks the sophistication of methods described in other similar published CUP classifier studies. Multi-class classification and machine learning methods are readily available including k-nearest neighbor, support vector machines or neural network. Typically, these classifiers are tuned using leave-one-out cross

validation on the training data and before testing on independent test set. Without this detail included in the manuscript the performance of the classifier cannot be trusted and neither can the results reported on CUP.

> We agree with the reviewer, which is why we have addressed this issue by setting up a validation cohort using 100 consecutive MASTER patients enrolled between 12/2020 and 06/2021. We only included entities that are part of TCGA to ensure comparability with other tissue-of-origin classifiers, which were usually trained on TCGA data and can therefore only reliably predict entities that are part of TCGA. We compared our results with two other published methods, cancerSCOPE (49 – 56 % accuracy) and CUP-AI-Dx (54% accuracy), both of which had similar accuracy as our entity prediction when using TCGA as a reference cohort (53% accuracy), but poorer accuracy than our entity prediction with MASTER as a reference cohort (75%). We have rewritten the respective result part of the manuscript and added the detailed validation cohort as supplementary information.

Furthermore, Zheng and colleagues published a DNA methylation-based deep neural network model with a very convincing accuracy (<https://doi.org/10.1371/journal.pone.0226461>). However, code documentation is insufficient to apply the trained algorithm to our MASTER cohort methylation data set.

4. Supporting histopathology data. It is stated that ESMO guidelines were used for selecting CUP cases but very little detail is provided about any immunohistochemistry staining done. A CK7 and CK20 staining profile should be presented if this data is available in addition to any other Dx stains used. Melanoma of unknown primary should be excluded from the CUP series.

> Available immunohistochemistry data was added as supplementary information in Supplementary Table 1.3. However, the diagnostic workup for CUP according to ESMO guidelines includes melanoma. Melanoma of unknown primary (MUP) is an entity that can and should be distinguished from melanoma with known primary. Therefore, we have not excluded MUP patients from our cohort since we took into consideration all cancers of unknown primary and not just carcinomas.

5. Virus DNA detection. This is an important molecular feature that can be detected by WGS and RNA-seq but has not been included.

> We agree with the reviewer and therefore, we have performed several virus detection methods based on DNA and RNA. The results and a corresponding method section have been added to the paper and its supplementary information. We used three computational approaches to detect viral infections from next-generation sequencing (NGS) data: a k-mer-based approach (Kraken2 version 2.1.2 [<https://doi.org/10.1186/s13059-019-1891-0>]), an assembly-based approach (P-DiP [<https://doi.org/10.1038/s41588-019-0558-9>]), and an alignment-based approach where the sequencing reads were aligned against concatenated assemblies of the human genome and all RefSeq viral genomes, in accordance with Arriba's workflow for the detection of viruses. To detect viral integration sites, we used Arriba version 2.1.0 for RNA-Seq data and VIRUSBreakend version 2.12.0 [<https://doi.org/10.1093/bioinformatics/btab343>] for DNA-Seq data. Three tumors were positive for human papillomavirus type 16 and another one was positive for human papillomavirus type 18. Integration of viral DNA into the host genome was detected in three cases inside/near the genes HOXA2, MIER1, SLC35D1, and TENM4. The respective result and method parts were updated accordingly.

6. Figure 1 should highlight platform used (WGS/WES).

> Figure 1 has been updated accordingly.

7. Figure 2. How does tSNE clustering relate to the two ToO classifier methods employed? How is the reader to interpret this data?

> We apologize that our figure legend did not explain the figure properly. We have updated the figure legend in order to clarify that Figure 2 A and B are a visualization of our methylome based predictions. While many TCGA entities show very distinctive clusters, some do not. The updated figure legend helps to interpret this data properly.

8. Fusions. Anecdotal reports of fusions in CUPs are interesting but specific fusions that reported anecdotally in other cancer types cannot be used as pathognomonic features. e.g. EZR-ERBB4 has only been reported in only one mucinous lung cancer therefore cannot be used as evidence to support a lung origin.

> We agree that a single molecular alteration is usually not sufficient to suggest a certain underlying entity. This is in line with our statement in the result section “Diagnostic interpretation of molecular profiling” in which we have stated “Not all of them correlated with entity prediction results based on transcriptome profiling and these alterations alone did not necessarily justify diagnostic reclassification. Still, together with omics-based entity predictions they offer meaningful information of diagnostic value and can be useful for further treatment decisions.” However, we have updated the result text to state clearly that the EZR-ERBB4 fusion has been reported in only one mucinous lung cancer case.

9. There appears to be some inconsistencies in the data relating to TMB. For instance, CUPs 16, 18, 21 and 51 are reported as hypermutated in Supp Table 6.1 based on > 100muts detected but in Figure 1 all cases have <10mut/Mb. How is the reader to compare and interpret these two TMB metrics? Cases with <10mut/Mb would not seem hypermutated.

> We apologize for not explaining this properly. There are different ways of measuring TMB (Table 1, <https://doi.org/10.1002/gcc.22733>). Historically, MASTER has used 100 SNVs + indels as cut-off for defining hypermutation. All patients in this cohort received molecular tumor board recommendations using this cut-off. Therefore, this is what we reported in the respective result part. However, over the last couple of years measurement using mutations/Mb has become widely used. Therefore, we used mutations/Mb for describing TMB in general in our cohort in Figure 1. We have updated the corresponding result and method section to describe this more clearly.

Reviewer 2

Reviewer #2, expert in German clinical trials (Remarks to the Author):

--

Möhrmann and colleagues report on comprehensive and integrative analyses on genomic, transcriptomic and methylomic level from a subpopulation of patients with CUP syndrome enrolled into the MASTER program. The analysis population comprises 70 patients.

Patients with CUP syndrome have a poor prognosis; several attempts have been made to improve outcome e.g. by identifying site of origin. Besides non-comparative studies, one randomized trial has been reported aiming to better classify CUP patients regarding tumor site of origin and recommend respective treatment, however, as also discussed by the authors, this trial could not show that this approach actually leads to improved patient survival. Another randomized trial (SHIVA trial) compared treatment and outcome of patients with exhausted treatments options (not CUP patients)

who received either standard of care or treatment as recommended by a molecular tumor board. The SHIVA trial was reported in 2016 and, unfortunately, could not show a substantial benefit for this approach either.

In brief, the manuscript adds to the existing body of evidence of translational diagnostics and treatment approaches for CUP syndrome or patients with exhausted treatment options. The overall number of patients is smaller compared to similar studies in the field and the patient population is likely not representative for CUP patients, because of the obvious bias towards younger patients. It provides some new aspects, and it reveals that such wealth and in-depth analyses can be contradictory and that there still remain many open questions who to translate such findings into patient care.

In the current version of the manuscript, there are several open issues that need to be addressed. This study is primarily a clinical study; thus, it should clearly report items as strongly recommended in reporting guidelines such as STROBE or REMARK. Especially regarding definition of analysis populations, calculation and reporting of clinical outcome parameters. I have the following specific comments that may help to improve clarity of the paper:

> We thank the reviewer for his valuable comments, which allowed us to improve our manuscript. We have improved our descriptions in order to clarify that we fulfill the reporting recommendations by guidelines like STROBE or REMARK.

1. Where there any general specific inclusion/exclusion criteria for the MASTER program? If yes, please report these criteria or provide them in the supplement. You refer to references 30 and 43, but this is not sufficient for the reader.

> We have added the following information under method section “NCT/DKTK MASTER”:
NCT/DKTK MASTER includes adults with advanced cancer across histologies who are younger than 51 years and patients with rare tumors, including rare subtypes of more common entities, regardless of age as cited in the manuscript. Patients must have exhausted curative treatment options and be in good general condition (Eastern Cooperative Oncology Group performance status of 0 or 1). Patients with cancers of unknown primary were included regardless of age due to its rarity.

2. Is the MASTER program registered with a clinical trial registry? If yes, please provide the respective registration identifier. If it was not registered, please provide the reason.

> We apologize for being unclear about this. NCT/DKTK MASTER is not a clinical trial and therefore not registered in a clinical trial registry. The MASTER program is a prospective multicenter observational study that provides a standardized diagnostic workflow which enables molecularly informed decisions for further therapy. Treatment recommendations are made in cooperation with treating oncologists following interdisciplinary discussion in a molecular tumor board. Depending on availability, further treatment could be given in clinical trials, off-label, or on-label. Within the reported cohort, none of the patients received targeted therapy within a registered clinical trial. This is partly because, unfortunately, the number of available clinical trials for CUP patients is very limited. We have updated the corresponding methods and result section accordingly.

3. Did you collect clinical patient data prospectively? You somehow describe this in section “Clinical and statistical analysis”, but this is very superficial. How have clinical data been captured? Was there a centrally managed EDC system? Please comment.

> Clinical patient data was collected prospectively. Within the MASTER program we have implemented a centrally managed electronic data capture system (ONKOSTAR), which we have used to capture, organize and manage all clinical data in our cohort. At the time of inclusion, clinical data from doctor`s letters, reports of imaging, histopathologic assessment and other diagnostic procedures were collected using a pre-specified data collection process. Afterwards, follow-up data on treatment and outcome of patients has been gathered for at least 24 months whenever possible. To address this more clearly, we have updated the corresponding method section accordingly.

4. What type of sample (core needle biopsy, resection of metastases) was required for participation in the MASTER program? Were analyses based on fresh biopsy at progression and / or at primary diagnosis? Please report on what types of samples you conducted the analyses.

> We used fresh frozen tumor specimens and matched normal control samples that were acquired at the time of inclusion in the MASTER program. Biopsies were obtained in the context of surgical procedures (25x resection) or by core needle biopsy (41x biopsy). Location of biopsy (as far as known) is listed in Supplementary Table 6. Supplementary Table 6 has been updated in order to address the question completely. Only in seven samples there were noteworthy exceptions as listed in Supplementary Table 6 (CUP-09: FFPE, CUP-14: bone marrow sample, CUP-17 and CUP-35: FFPE and already delivered as DNA extract, CUP-36: lumbar puncture sample; CUP-48: macrodissection for tumor cell enrichment; CUP-59: FFPE). The corresponding method part has been updated accordingly.

5. Please provide a statement on the type of design of your study. Given the information provided, I anticipate this is an observational prospective cohort study, but this needs clear statement in the paper.

> We have included the following part in the method section under “NCT/DTKT MASTER”:
“NCT/DTKT MASTER is a prospective, continuously recruiting, multicenter observational study.”

6. I cannot find any section commenting on written informed consent of patients and ethical review of the study. Especially regarding the germline analysis, patients usually have to provide additional written consent. Please add this to the manuscript.

> We have included the following part in the method section under “NCT/DTKT MASTER”:
“Patients provided written informed consent for banking of tumor and control tissue, molecular analysis including germline analysis, and the collection of clinical data under a protocol (S-206/2011) approved by the Ethics Committee of the Medical Faculty of Heidelberg University. The study was conducted in accordance with the Declaration of Helsinki.”

7. You obviously used the program R for the bioinformatical part, but I cannot find a description of the statistical program you used to calculate the clinical outcome parameters such as PFS and response. Please comment on this and provide description in the methods section.

> Since our clinical analysis relied on basic mathematical calculations, we used Microsoft Excel and the R script ggplot2 to facilitate this process. We prepared tables similar to our supplementary tables to organize clinical data and calculate relevant parameters such as PFS. We have added the following phrase under the method section “Additional data processing and analysis”:
“PFS, PFSr and mPFSr were calculated using Microsoft Excel. Survival analysis using Kaplan-Meier estimator and log-rank tests was performed using ggplot2 (version 3.3.3). P values < 0.05 were considered statistically significant.” Minor mistakes were corrected in the manuscript.

8. It is not clear to me whether reported patients were included from one or several centers. You mention the DTKT consortium, but this is not really clear for readers not being familiar with current structures in Germany. Was this a single or multicenter study? If multicenter, from how many centers were patients included?

> All DTKT sites are actively recruiting patients within the MASTER program. We have added the following information under the method section “Clinical and statistical analysis”:
“Clinical data for cohort selection, description and analysis was obtained from the National Center for Tumor Diseases (NCT) Heidelberg and Dresden as well as from six other comprehensive cancer centers (CCCs) of the German Cancer Consortium (DKTK). The DTKT network includes ten CCCs at eight sites (Berlin, Dresden, Essen/Düsseldorf, Frankfurt/Mainz, Freiburg, Heidelberg, Munich, Tübingen).”

9. Where have the analyses been conducted? Was it one laboratory or several sites? If yes, how was standardization of assays etc. guaranteed?

> All molecular analyses were done centrally in Heidelberg using the well-established and standardized clinical genomics workflow of the MASTER program. All samples underwent processing by the NCT Molecular Diagnostics Program Sample Processing Laboratory (SPL) and sequencing analysis by the DKFZ Genomics and Proteomics Core Facility (GPCF). Detailed information concerning test kits and quality control is listed in the method section as well as in the supplements. We have updated the corresponding method section accordingly.

10. Did you pre specify tests and panels to be done? E.g. a minimal core set of WES followed by other techniques / analyses if sufficient tissue was available?

> From May 2013 to October 2016 whole exome sequencing (WES) was the standard sequencing method within the MASTER program. Since November 2016 we performed whole genome sequencing (WGS) instead of WES in a substantial proportion of patients. If RNA quality was sufficient, RNAseq was performed as described in the manuscript. If tumor cell content was >30%, methylation analysis was performed as described in the manuscript but methylome information was not available for the molecular tumor board since it was generated later than DNA-/RNAseq data. We have described this more clearly in the manuscript and updated Supplementary Table S15.

11. Regarding the entity prediction, were results from these analyses available to the MTB when discussing the cases and making recommendations?

> We apologize for not being precise enough when describing this part in the manuscript. Entity predictions were calculated retrospectively and therefore not available for treatment recommendations by our molecular tumor board. In the result section “Entity prediction using methylome and transcriptome data” we have added the sentence:

“We retrospectively performed entity predictions using methylome- and transcriptome-based similarity analysis.”

In the discussion we have added the sentence:

“In our cohort, MTB recommendations were not influenced by our methylome- and transcriptome-based entity prediction since it was not available at the time of the MTB.”

12. The reported 70 patients were relatively young and are likely not representative for patients usually diagnosed and treated with CUP syndrome. How do you explain this? May this be because of your inclusion/exclusion criteria?

> We agree with the reviewer, as also addressed in our discussion; the reported patients were younger than we would expect them to be in a representative cohort based in Germany (Kraywinkel und Zeissig, 2017; abstract available in English, <https://doi.org/10.1007/s00761-017-0301-z>). NCT/DKTK MASTER includes patients between age 18 and 50 and/or patients with rare cancer types such as CUP regardless of age. This might have influenced oncologists to particularly refer younger CUP patients to our program. Additionally, we only included patients who had exhausted curative treatment options and were in good general condition (Eastern Cooperative Oncology Group performance status of 0 or 1). Younger patients are more likely to fulfil these criteria. We have updated the corresponding discussion part of the manuscript.

13. You report to have identified and included 70 patients with CUP syndrome, however, only 61 patients fulfilled the ESMO criteria. I assume this is caused by discrepancy between inclusion criteria of the MASTER program and the ESMO criteria. Why is this? Please comment on this.

> As stated in the result part of the manuscript, the documentation of the initial imaging procedures (such as CT scans of thorax, abdomen and pelvis) was not available in 9 patients. According to ESMO guidelines these are required for the diagnosis of CUP syndrome. Nevertheless, these patients were enrolled in the MASTER program since we relied on the diagnosis of the referring treating oncologists. We have updated the corresponding method section to address this more clearly.

14. Please provide percentage in addition to the absolute numbers in table 1.

> We have provided an updated table with percentages.

15. One of most central major revisions required is the definition of analysis populations and outcome reporting. Current presentation of clinical outcome is certainly biased.

> We refer to our answer to comment 16.

16. Specifically: First, all 70 patients should be in the denominator when reporting outcome, thus, for all 70 patients PFS and OS should be calculated and reported. Also, for all these 70 patients you should calculate the PFS ratio irrespective of the MTB recommendation or whether recommended treatment has been given. Second, if you want to explore potential associations of benefit between personalized treatment and outcome, you need at least to contrast outcome (PFS, response, PFS ratio, OS) of patients who received the recommended treatment (N=20), but also the outcome of those patients who had a recommendation but did not receive the recommended treatment (N=36). A stratified table, similar to content of table 1, would be helpful allowing to compare patient and disease characteristics. In addition, information on the evidence level of the MTB recommendation would also enrich the content. However, such exploratory analysis would still not allow to conclude that personalized treatment based on an MTB leads to improved patient outcome, because of several unmeasured confounding factors. For such statement you need a randomized trial. This limitation also needs to be addressed in the discussion.

> We thank the reviewer for these important remarks. It is true that our study is not a randomized trial and we listed this topic in the revised manuscript more clearly among the limitations of our study that we have already addressed in the discussion.

We followed the approach of PFS2/1 ratios in patients receiving genomics-based therapies that was initially described by Von Hoff et al. J Clin Oncol 2010 (<https://doi.org/10.1200/jco.2009.26.5983>). In this analysis each patient is his/her own control comparing the molecularly informed treatment (PFS2) with the last systemic treatment before molecular analysis (PFS1). This approach has also been used in various published molecular profiling studies such as the MOSCATO or WINTHER trial which are cited in the manuscript. Nevertheless, we have added the clinical outcome of patients that did not receive a recommended targeted treatment as proposed by the reviewer. We have added the corresponding PFS as well as OS and response data of this cohort as requested. We have divided the clinical results into the chapters "Genomics-based treatment recommendations" and "Genomics-based systemic treatment" in order to keep it well-arranged for the reader.

Furthermore, we provide a stratified table as requested (Supplementary Table S13). We included NCT/DKTK evidence levels and further outcome parameters in the corresponding result sections: "All drug recommendations were sorted into groups with the evidence level they were based on (Level 1 A/B/C, 11/142, 8%; Level 2 A/B/C, 89/142, 63%; Level 3, 31/142, 22%; Level 4: 11/142, 8%)." "The distribution of evidence levels of clinically applied recommendations showed a similar distribution as the one of all drug recommendations (Level 1, 2/20, 10%; Level 2, 15/20, 75%; Level 3, 1/20, 5%; Level 4, 2/20, 10%)."

"Of 20 patients who received the recommended targeted therapies, 12 (60%) had stable disease \geq 6 months or achieved objective remissions (PR/CR)."

"For patients that did not receive a recommended treatment, we calculated the ratio of the first treatment applied after the MTB (PFSb, n = 12) and the last prior systemic treatment (PFSa, n = 11) which resulted in a mean PFSr of 0.67 (median PFSr = 0.71, range 0.1 to 1.0, n = 11; Supplementary Table S14). Median overall survival of the 36 patients without application of recommended treatments was significantly shorter than of the 20 patients that received a recommended therapy (18.3 months vs. 34.8 months, p < 0.05; Supplementary Figure S4). Same was true for median PFS2 and PFSb (7.8 months vs. 3.8 months, p < 0.001; Supplementary Figure S4). Two patients with PFSb had stable disease \geq 6 months or achieved objective remissions (PR/CR; Supplementary Figure S4).

Since our study was not randomized, these results are not controlled for possible confounding factors. Further data are provided in Supplementary Table S13 and S14.”

In the discussion we have added the sentence:

“Third, our study was not a randomized clinical trial but a prospective observational study.”

17. For all Kaplan-Meier plots, please provide 95% confidence intervals and at-risk table below the graph as this is standard in reporting time-to-event data.

> We have updated all Kaplan-Meier plots providing 95% confidence intervals and number at risk tables below the graph (Supplementary Figure S4).

18. Please report the absolute number of PFS defining events together with PFS. Please also report OS and the absolute number of deaths. Finally, also report the follow-up time of all patients, e.g. using the inverse Kaplan-Meier method.

> Of 20 patients that received molecularly-guided therapy, 19 showed progressive disease and one death as PFS defining event. The corresponding result and method section have been updated. The absolute number of deaths during the observation period in the whole cohort of 70 patients was 38, we have introduced this information in the result section “Patients”. Median OS of those 38 patients is 12.0 months as reported in supplementary table 1. Follow-up time of all patients using the inverse Kaplan-Meier Method has been added in Supplementary Figure S4 as suggested by the reviewer. Median follow-up time (25.2 months) and median OS (22.1 months) have been added to the result section “Patients”.

19. How was treatment in accordance with the MTB recommendation defined? Please specify this.

> Only systemic treatments that matched with at least one of our molecular tumor board recommendations for the given patient and that were initiated after inclusion into the MASTER program were considered as treatment in accordance with MTB recommendations. We have updated the method section with the sentence:
“Progression-free survival (PFS) of the first applied treatment recommended by the MTB (PFS2) was compared to the PFS of the last prior systemic treatment (PFS1) in each individual patient.”

20. Where there any pre-specified assessments or where patient outcomes determined based on routinely collected data? Please specify.

> Patient outcomes were determined by centrally and routinely collected clinical data based on doctors’ letters and diagnostic reports including histopathological assessments or imaging procedures at baseline and during follow-up. All clinical data were reviewed and assessed by a team of medical curators on the basis of pre-specified rules for a harmonized interpretation. We have updated the corresponding method section in order to describe this more clearly. We are aware of the limitations of observational studies. Nevertheless, our observational data provide valuable and clinically important real-world data on the impact of genomics-guided therapies in CUP patients and may pave the way for corresponding clinical trials to confirm the efficacy of molecularly stratified treatment approaches in this entity.

21. How was PFS calculated? Please provide the starting date for calculating the PFS as well as definition of PFS defining events.

> In each individual case, the starting date of a given PFS was the first day of drug application. Due to data protection guidelines we are not allowed to provide individual starting dates in the manuscript. End of PFS was either death or progressive disease. We have updated that definition in the manuscript.

22. How did you collect information PFS of the most previous therapy before inclusion into the MASTER program?

> PFS1 was assessed the same way as described in our answer to question 21. PFS started with the first day of therapy application and ended with documented disease progression. As mentioned above, all data were reviewed by a team of curators.

23. Have you collected the reasons why recommended treatments were not applied? Was this a reimbursement issue? This is a very central issue in all initiatives of personalized oncology treatment. Please comment.

> Within our MASTER trial, reasons for non-implementation of MTB recommendations included, e.g. worsening of patient's general condition, death before treatment could be given, and lack of access to or reimbursement of the recommended drug(s) as described in <https://doi.org/10.1158/2159-8290.cd-21-0126>. We have updated the corresponding result part with this information. More detailed information has been introduced in Supplementary Table 14.

24. Please report all outcomes in months, not in days.

> We converted all outcome data to months as requested by the reviewer (please see manuscript and Supplementary Table S12 in particular).

NCOMMS-21-08986B (minor revision)

Reviewer #1 (Remarks to the Author):

This is a second review of the manuscript prepared by Möhrmann and colleagues. I think the manuscript has been improved and I praise the authors for the extra effort. I can accept most of the responses made to my initial review questions. However, I do feel there are some parts that could still be improved further to better describe the data and analysis done.

> We thank the reviewer for his positive response to our revised manuscript. In the following, we would like to address each comment point by point.

Molecular characteristics

1. May I suggest the authors consider adding some further detail in Figure 1 where new information has come to light as this is relevant to the molecular landscape of the CUP cohort as well as diagnosis and treatment decisions made

- Consider replacing “HRD” with “LOH–HRD + LST” in the Figure to differentiate from the other methods used to infer HRD. Also consider adding annotation for CHORD and HRDetect where the score was above the threshold prescribed for calling HRD in the original papers describing these methods.

> We have added this information to Figure 1 as requested.

- It would also be useful to add any data relating to mutated HRD-related genes (e.g BRCA1, CHEK2)

> We introduced an additional annotation bar in the Figure 1, which shows the presence of SNVs, indels or fusions in HRD-related genes. To do so, we queried 12 HRD-related genes (Supplementary Table S17.1) in which mutations were previously observed [Mateo et al. 2015, NEJM, DOI: 10.1056/NEJMoa1506859]. Details about mutations underlying the annotation were included in an additional supplementary table.

- Add dominant mutational signatures AC4 and AC7 to the Figure.

> We have added this information to Figure 1 as requested.

- Add annotation where HPV sequence was detected as these have important diagnostic relevance.

> We have added this information to Figure 1 as requested.

- The authors could consider adding a histology annotation key in the figure. For instance, one might like to know if HPV was detected in SCCs.

> We have added this information to Figure 1 as requested.

- Include data for cancer genes associated with homozygous deletion (e.g. CDKN2A n=7) as this is an important feature like fusion events, SNVs and InDels. HD of CDKN2A is mentioned even in the abstract so it seems appropriate to include in the Figure?

> In our study, we identified focal somatic copy number alterations in CUP patients. We agree with the reviewer that the homozygosity of deletions is interesting and could be presented in our already comprehensive Figure 1. In our analysis, we define focal event as a distinct copy-number segment spanning less than 3 Mb. Such a threshold allows for recognition of almost all homozygous deletions [Cheng et al. 2017, <https://www.nature.com/articles/s41467-017-01355-0>]. However, their reliable detection can be challenging. The two pipelines that we use for CNV analysis, due to the nature of the data (WES and WGS), perform differently in that respect causing under-calling (CNVkit) or over-calling (ACEseq) of homozygous deletion events. To overcome the over-calling problem, we focus on genes that are relevant to cancer, namely those present in Cancer Gene Census. We added these results to Figure 1. In short, the most frequent (n>1) focal homozygous deletions observed in Cancer Gene Census genes are: CDKN2A (n=6), FHIT (n=4), ROBO2 (n=3), CCR7 (n=2), FOXP1 (n=2), LRP1B

(n=2). In the previous version of manuscript, we wrote that we observe focal deletion of *CDKN2A* in seven patients, with five being homozygous and two - hemizygous. After deeper look into the results we reclassified one hemizygous patient to the homozygous group, because although most part of *CDKN2A* gene in this patient is indeed a hemizygous deletion, there is a small 230 bp long segment that is classified as homozygous deletion. Therefore, we modified the text accordingly. Moreover, we have improved the clarity of our description: adding a word “homozygous” in the abstract and introducing the definition of focal copy number alteration in the methods.

Germline analysis

2. State whether somatic LOH was concordant with pathogenic germline mutations. Germline mutations with concordant LOH could also be added to Fig 1 as above.

> The only patient with a (likely) pathogenic germline variant and concordant LOH is CUP-64 who has a *CDKN2A* germline variant. Since it is only one patient, we did not add this information to Figure 1 but introduced the information in the corresponding result part in the manuscript. We added LOH annotation in the respective supplementary table.

3. Line 214. There is an incorrect callout of Supplementary Table 4.1 (should be Supp Table 5.1)

> We have verified that all tables and figures are referred to correctly in the manuscript. While doing that we could not confirm an incorrect callout in line 214.

4. I note that the heterozygous germline FH variant c.1431_1433dupAAA has recently been shown not to be not associated with increased risk of developing fumarate deficient HLRCC

<https://pubmed.ncbi.nlm.nih.gov/31444830/>

> We thank the reviewer for bringing this to our attention. We agree that this variant should rather be assessed as variant of unknown significance in regards to cancer predisposition. Therefore, we have updated the assessment in the respective supplementary table. Furthermore, we have changed the description of our 101 gene list used for filtering of rare germline variants to “cancer predisposition genes” in the text as well as in the supplement in order to be more concise.

Gene-expression and DNA Methylation Classification

5. The authors have now tested a modest number of tumors from the MASTER set using the TCGA trained classifiers. Previous studies have shown that gene-expression and DNA methylation tissue of origin classifiers can achieve test accuracies for metastatic tumors in the order of 80-90%. Could it be that the poorer performance (~49-56%) observed using the three TCGA trained RNA classifiers (incl. CUP-AI-Dx, CancerScope and the authors own method) relates to a systematic bias existing between MASTER and TCGA datasets? For instance, the TCGA set is made of primary cancers, whereas the gene-expression classifiers are now being tested on metastatic cancers? This caveat should be acknowledged in interpreting the results and making any conclusions about the relative performance of these methods on the MASTER CUP cohort.

> The reasons for the relatively poor performance are multifactorial. The overrepresentation of metastatic disease in the validation cohort and the lack thereof in the TCGA reference cohort are probably an important factor, as suggested by the reviewer. Another likely reason is that the MASTER cohort by design comprises rare cancer types, thus impairing the match-making between MASTER and TCGA patients. Moreover, we note that a number of misclassifications concern cancer types with transcriptionally similar cells of origin. For example, frequent misclassifications were made between cholangiocellular (CHOL), pancreatic (PAAD), and hepatic cancer (LIHC), presumably owing to a common site of metastasis and possibly also transcriptionally similar cells of origin. Likewise, uveal melanoma (UVM) and skin cutaneous melanoma (SKCM) were also swapped in several cases. While these lesions may be straightforward to distinguish using external information (e.g., anatomical site of lesion), the classification exclusively based on transcriptomic data is challenging. Lastly, technical differences in the library preparation protocols and sequencing protocols of MASTER and TCGA certainly contribute to the misclassification rate as well, since expression quantification using RNA-

Seq is known to be sensitive to such protocol differences. We have incorporated these thoughts into the discussion.

6. Line 260 Written “This approach showed the best accuracy when tested on our validation cohort (54/72, 75%) and enabled us to have a comparison to rare tumor entities which were enrolled in MASTER but are not part of TCGA”. Please provide more information on these additional rare cancer types represented in the MASTER series and not in TCGA. The authors may also need to consider whether the reference transcriptome data for the MASTER cohort needs to be made available for reproduction of the classification results. Would this not be a requirement of the journal? I tried accessing the EGA accession cited in the manuscript but I could not see what data has been uploaded.

> MASTER particularly focuses on young patients and patients with rare cancers. Therefore, MASTER includes entities that are not part of TCGA. Examples are DSRCTs, adenoid cystic carcinomas and several types of sarcomas. We have added a complete list of diagnoses that are part of the reference cohort in an additional supplemental table. Additionally, we have uploaded the expression values of the reference cohort and the validation cohort as a matrix of FPKM values to EGA. Moreover, the methylation beta values of the validation cohort have been added as well.

7. Regarding the method used for RNA classification. How did the authors arrive at the arbitrary threshold of FPKM<>5 for the RNA classifier? Was there any tuning using a leave one out cross validation of the training dataset? As per my in comments from initial review a LOOCV and tuning of a classifier is standard practice as it functions to tune the classifier but also show the relative performance can be maintained between training and independent test sets.

> The thresholds were defined based on biological rationale. The gist of the algorithm is the selection of marker genes that distinguish samples/tissues from each other. To this end, genes are selected which are expressed in one tissue but not in the other. A threshold of <0.5 FPKM was assumed to select genes that are not expressed, and a ten-fold higher threshold (>5 FPKM) was assumed to be sufficiently larger to select stably expressed genes.

We followed the reviewer’s advice and performed 10-fold cross-validation on a subset of the reference cohort, which served as a training set. The following heat map shows the prediction accuracies across the explored parameter space. We evaluated all integer combinations between 0 and 20 FPKM for the two thresholds for non-expressed and highly expressed genes. The procedure identified the combination of 3 FPKM for non-expressed genes and 13 FPKM for highly expressed genes to be the optimum.

Next, we applied the optimized parameters to the validation cohort, which confirmed that the new parameters indeed improve the predictions (three more correct predictions than with the non-optimized parameters).

Therefore, we have updated the CUP predictions in the manuscript based on the new thresholds.

8. Line 249-250 Written “When comparing TCGA cohorts with our CUP cohort, classification based on methylome comparison led to an entity prediction in 55/70 cases (79%).” I find this sentence a little misleading as it suggests that the classifier could only make a prediction in 55 cases but really what is meant is that prediction was only attempted in 55 cases. I feel this section could be written in a more concise manner.

> We have rewritten this sentence, the wording has been changed to “classification ... was possible.”

9. Line 264 Written “It highlights the need for an integrated classifier taking into account both methylation and transcriptomic data (Figure 2)”. I think the description now provided in the legend for Figure 2 including a justification for the tSNE plots and the interpretation of the data should be in the main body of the text. The authors should also perhaps reconsider the call out of tSNE plots in panels 2A and 2B. One suggestion is to call out (2A, 2B) at the very beginning of the Results section titled “Entity prediction using methylome and transcriptome data”. Something general could be said about the heterogeneity of the CUPs based on DNA methylation (GEP) profile, that CUPs cluster among a range of solid cancers or carcinomas but to the exclusion of some cancer types. This could then be a segue to describing the multi-class classification methods. I think for completeness a tSNE or UMAP of transcriptome data should also be shown for comparison to methylation. I will also add that other than the reader getting the impression from tSNEs that CUPs are heterogeneous and cluster among a range of cancer types the plots are rather impossible to interpret given the large number of cancer types and spectrum of colors in the cancer type key. For instance, the comment that CUPs cluster with PAAD and CHOL is hard to see in the plot. Perhaps the

authors could consider improving this figure by embedding class labels proximal to clusters within the tSNE itself or at least highlighting some of the major tumor clusters within the plot so the reader can easily interpret the data.

> We have added tSNE plots based on the transcriptome data as requested (Figure 2C+D). Additionally, we have also rearranged the manuscript and put part of the figure legend into the result section as requested by the reviewer.

However, the suggested embedding of class labels creates an even more complex plot and most of the labels in the middle of the methylome plot highly overlap (for illustration see eclipse representation of regions by entity in the tSNE plot attached). For this reason, it will not be possible to highlight just some clusters to better interpret the CUP samples within the methylome plot. We have tried this before but there is no way that by means of a tSNE plot one can easily and correctly assign the individual CUP points to entity clusters just by eyeballing. The same problem occurs with the transcriptome tSNE plot although the overlap is slightly less severe. We would further argue that this is not the purpose of the tSNE plot. The reader who is interested in the detailed stats can refer to the supplemental tables that contain helpful information. Figure 2A-D is intended to only illustrate the heterogeneity of CUP samples and which entity clusters they roughly align to in these two dimensions.

10. Panel 2C. Total cases n=55 is shown in the non-overlapping portion of the Venn circles for all classifiers. My understanding is that the convention in a Venn diagram would be to show the number of non-concordant cases in the outer non-overlapping part of the circle. Furthermore, the number of cases in the direct comparison between gene-expression and methylation is 48 not 55 as written. Therefore the number in the non-overlapping part of the circle is 48 minus the number of cases overlapping with either of the other two classification methods. A separate panel should be made for direct comparison of the gene-expression classifiers as the number of cases tested is different.

> We have included a new version of figure 2C (now called Figure 2E). As suggested, we corrected the number of total comparable cases to 48 and replaced the case-counts inside non-overlapping circles

with respective numbers of non-concordant cases. Furthermore, we have provided a separate panel for direct comparison of the gene-expression classifiers as proposed by the reviewer (Figure 2F).

Reviewer #2 (Remarks to the Author):

The authors have sufficiently addressed my previous comments. I still have some final suggestions:

Regarding “Statement of Significance”, the conclusions are too strong regarding the clinical benefit. I strongly suggest tuning down the wording, e.g. “...targeted therapies showed some clinical activity, but definitive clinical benefit of this approach requires further evaluation in dedicated clinical trials.”

> We have updated the wording according to the reviewer’s wishes.

“Statement of Significance” - old version:

Treatment options for CUP patients are limited and often ineffective resulting in a dismal prognosis. We demonstrate that a comprehensive precision oncology approach provides clinically relevant information and additional, molecularly stratified treatment options in many cases. The majority of these targeted therapies was highly beneficial even in heavily pretreated patients.

“Statement of Significance” - new version:

Treatment options for CUP patients are limited and often ineffective resulting in a dismal prognosis. We demonstrate that a comprehensive precision oncology approach provides clinically relevant information and additional, molecularly stratified treatment options. Targeted therapies showed clinical activity even in pretreated patients warranting further evaluation in dedicated clinical trials.

In the chapter “Genomic based treatment recommendations”, please also mention here that only sequencing data were available to MTB when discussing patients and making treatment recommendations.

> We have introduced this information in the first sentence of the respective result chapter as recommended by the reviewer.

Please also provide the turn-around times from testing to first discussion of the patients at the MTB. This time (median, range) should also be shown in table S13, stratified and in total.

> We have added the turn-around times from testing to MTB in Supplementary Table 1 and mention the median value (and range) in the result section as well as in the corresponding supplementary table as requested.

Table S13: The variable “MTB recommendations” does not sum up to the denominator of 56 patients; there should not be two units of investigations in one table. Also, please provide a total column for the column.

> We apologize for this mistake. We have updated the supplementary table as requested.

REVIEWERS' COMMENTS

Reviewer #1 (Remarks to the Author):

The authors have now adequately addressed all of my comments

Reviewer #2 (Remarks to the Author):

The authors have sufficiently answered/edited all my requests, no more comments.